# Transformers and capsule networks *vs* classical ML on clinical data for alzheimer classification

Mario Alejandro Bravo-Ortíz[1,2,3], Sergio Alejandro Holguin-Garcia[1,2,3], Ernesto Guevara-Navarro[1,2], Esteban Cerón-Cabrera[2], Alejandro Mora-Rubio[2], Harold Brayan Arteaga-Arteaga[2], Gonzalo A. Ruz[4,5,6,7] and Reinel Tabares-Soto[1,2,4,8]

[1] Departamento de Sistemas e Informática, Universidad de Caldas, Manizales, Caldas, Colombia
[2] Departamento de electronica y automatización, Universidad Autónoma de Manizales, Manizales, Caldas, Colombia
[3] Centro de Bioinformática y Biología Computacional (BIOS), Manizales, Caldas, Colombia
[4] Facultad de Ingeniería y Ciencias, Universidad Adolfo Ibáñez, Santiago, Santiago, Chile
[5] Millennium Nucleus for Social Data Science (SODAS), Santiago, Chile
[6] Center of Applied Ecology and Sustainability (CAPES), Santiago, Chile
[7] Data Observatory Foundation, Santiago, Chile
[8] GobLab Escuela de Gobierno, Universidad Adolfo Ibáñez, Santiago, Chile

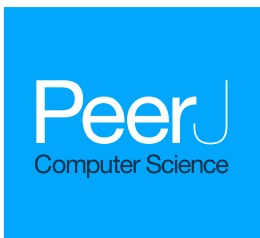

Corresponding author
Mario Alejandro Bravo-Ortíz,
mario.bravoo@autonoma.edu.co

## ABSTRACT

Alzheimer's disease (AD) is a progressive neurodegenerative disorder and the leading cause of dementia worldwide. Although clinical examinations and neuroimaging are considered the diagnostic gold standard, their high cost, lengthy acquisition times, and limited accessibility underscore the need for alternative approaches. This study presents a rigorous comparative analysis of traditional machine learning (ML) algorithms and advanced deep learning (DL) architectures that that rely solely on structured clinical data, enabling early, scalable AD detection. We propose a novel hybrid model that integrates a convolutional neural networks (CNNs), DigitCapsule-Net, and a Transformer encoder to classify four disease stages—cognitively normal (CN), early mild cognitive impairment (EMCI), late mild cognitive impairment (LMCI), and AD. Feature selection was carried out on the ADNI cohort with the Boruta algorithm, Elastic Net regularization, and information-gain ranking. To address class imbalance, we applied three oversampling techniques: synthetic minority oversampling technique (SMOTE), oversample using adaptive synthetic (ADASYN), and SMOTE-Tomek. In the three-class setting, the CNN + DigitCapsule-Net hybrid attained 90.58% accuracy, outperforming state-of-the-art baselines that rely only on clinical variables. A tuned gradient boosting (GB) model achieved comparable performance with substantially lower computational requirements. Model interpretability was assessed with SHAP and gradient-weighted class activation map (Grad-CAM), which identified Clinical Dementia Rating-Sum of Boxes (CRD-SB), Logical Memory-Delayed Recall Total Number of Story Units Recalled (LDELTOTAL), and Modified Preclinical Alzheimer Cognitive Composite with Trails B (mPACC-TrailsB) as the most informative clinical features. This combination of predictive strength, computational efficiency, and transparent interpretation positions the proposed approach as a promising open-source tool for facilitating early AD diagnosis in clinical settings.

## INTRODUCTION

Alzheimer's disease is a neurodegenerative disorder that initially affects the hippocampus and adjacent cortical regions—particularly the frontal and temporal lobes—before progressing to the neocortex. The rate of progression varies among individuals. AD is characterized chiefly by the accumulation of insoluble $\beta$-amyloid peptides that form extracellular plaques and vascular deposits (*Masters et al., 2015*). It remains the most common form of dementia and predominantly affects individuals older than 65 years (*Jo, Nho & Saykin, 2019*). Demographic projections indicate that adults aged $\geq 65$ years will constitute 17% of the global population by 2050, resulting in an estimated 152 million AD cases and raising major public-health concerns in the absence of curative treatments (*Uwishema et al., 2022*).

AD progresses through distinct and well-characterized clinical stages. According to the Alzheimer's Disease Neuroimaging Initiative (ADNI; https://adni.loni.usc.edu), the clinical stages considered in this study are:

- **Cognitively normal (CN):** Individuals exhibit no measurable cognitive impairment and serve as healthy controls.
- **Early mild cognitive impairment (EMCI):** Individuals remain socially engaged and functionally independent yet begin to experience episodic memory lapses, word-finding difficulties, or frequent misplacement of familiar objects.
- **Late mild cognitive impairment (LMCI):** Functional abilities decline further; individuals may require assistance with daily tasks, experience communication difficulties, and exhibit behavioural or personality changes, including fine-motor impairment.
- **Alzheimer's disease (AD):** Patients demonstrate severe cognitive and functional impairment requiring full-time care. Common symptoms include profound disorientation, significant behavioral disturbances such as aggression, and an inability to recognize close family members.

AD remains difficult to diagnose, prompting the development of diverse investigative strategies, including fluid-based and neuroimaging biomarkers (*Dubois et al., 2021*; *Muksimova et al., 2025*). In recent years, deep learning (DL) methods have advanced rapidly (*Muksimova et al., 2024*); convolutional neural network (CNN) and transformer-based architectures trained on magnetic resonance imaging (MRI) data now achieve high staging accuracy (*Mora-Rubio et al., 2023*). Notable models include CapsuleNet, which replaces standard convolutional layers with hierarchical "capsules." When coupled with dynamic routing, these capsules preserve spatial hierarchies, detect salient features, and limit capsule-vector magnitudes through a squashing nonlinearity

(*Liu, Li & Li, 2020*). Transformers represent another advanced approach; self-attention enables them to focus on diagnostically relevant MRI regions or characteristic temporal patterns in electroencephalography (EEG) data, thereby improving AD-stage classification accuracy (*Miltiadous et al., 2023*; *Bravo-Ortíz et al., 2024b*).

Nevertheless, structured clinical data—such as laboratory results, cognitive-test scores, and family history—remain underutilized in AD research. Common assessments include the AD Assessment Scale Cognitive Subscale (ADAS-Cog), the Rey Auditory Verbal Learning Test (RAVLT), the Mini-Mental State Examination (MMSE), the Functional Assessment Questionnaire (FAQ), and the Trail Making Test Part B (TMT-B) (*Battista, Salvatore & Castiglioni, 2017*). The presence of the apolipoprotein E ε4 (APOE ε4) allele, detected in more than 50% of patients and linked to early disease onset, $\beta$-amyloid deposition, and tau neurofibrillary tangle formation, is a well-established genetic risk factor yet remains comparatively underexplored (*Michaelson, 2014*; *Tan et al., 2021*). The present work offers a comprehensive comparison of seven models—CNN + DigitCapsule-Net, CNN + transformer encoder (CNN + TF), stand-alone CNN, Extra-Trees Classifier (ETC), support vector classifier (SVC), Random Forest (RF), and gradient boosting (GB)—to classify the AD, CN, EMCI, and LMCI cohorts using exclusively clinical variables. Rigorous hyperparameter optimization and architectural refinement were conducted to maximize classification accuracy. Model interpretability was evaluated with gradient-weighted class activation mapping (Grad-CAM), Shapley additive explanations (SHAP), and feature-importance analyses to elucidate the contributions of individual clinical variables to model decisions. The main contributions of this article are summarized as follows:

- This study presents structured clinical data, including laboratory values, biomarkers, and family history for AD detection, providing a more accessible and cost-effective alternative to imaging-based methods.
- A comprehensive evaluation of traditional ML algorithms and state-of-the-art DL models reveals that well-optimized ML techniques can outperform advanced DL architectures in accuracy and stability when applied to clinical variables.
- Novel hybrid architectures that integrate convolutional feature extractors with DigitCapsule-Net or Transformer encoders are proposed and assessed. These models significantly improve classification performance across CN, EMCI, LMCI, and AD cohorts, highlighting the potential of clinical-data–driven diagnostics.
- Reproducibility is ensured by releasing all datasets and source code, enabling independent validation and fostering continued innovation in artificial intelligence (AI)-based AD diagnosis; the release also clarifies the strengths and limitations of clinical-data-driven methodologies.

The remainder of the article is organized as follows. 'Related Work' reviews related work; 'Materials and Methods' describes the materials and methods, including the clinical dataset, preprocessing pipeline, class-balancing techniques, model architectures,

evaluation metrics, interpretability tools, and computational setup. 'Results' presents the experimental results, which are discussed in 'Discussion', and 'Conclusion' concludes.

## RELATED WORK

Projections indicate that AD—already the leading cause of dementia—will become one of the world's most lethal disorders, with epidemiological models predicting that global dementia prevalence will triple by 2050 (*Scheltens et al., 2016*). Historically, AD diagnosis required the presence of dementia—a progressive syndrome of substantial cognitive decline that severely impairs daily functioning. Early detection is therefore essential for delaying progression to this debilitating stage. According to *McKhann et al. (2011)*, standardizing AD classification remains challenging because many diagnostic criteria are overly broad. The diagnostic criteria are summarized below:

1. The histopathological features of Alzheimer's disease manifest across a broad clinical spectrum.
2. Distinctive characteristics that differentiate AD from other dementias in comparable populations remain poorly defined.
3. Although memory impairment is often the earliest symptom, non-amnestic presentations are also observed.
4. The age distribution of the population vulnerable to Alzheimer's disease has shifted over time.
5. Genetic determinants of Alzheimer's disease are still not fully elucidated.

The National Institute on Aging classifies Alzheimer's-related dementia into three clinical subtypes:

1. **Probable Alzheimer's dementia**

    (a) Meets the National Institute on Aging diagnostic criteria for dementia (*McKhann et al., 2011*).
    (b) Exhibits cognitive deficits, including memory loss and impaired learning.
    (c) Demonstrates difficulty with face and object recognition, word-finding, reasoning, and judgment.

2. **Possible Alzheimer's dementia**

    (a) Meets the above criteria but presents with an abrupt onset of cognitive decline.
    (b) Shows clinical or imaging evidence of cerebrovascular disease.

3. **Probable Alzheimer's dementia with evidence of pathophysiological processes**

    (a) Biomarkers indicate neuronal injury or degeneration.

Current diagnostic criteria integrate neuropsychological evaluations and standardized cognitive assessments, whereas researchers typically investigate neuronal degeneration

with biomarker-based MRI. Although clinical criteria provide valuable insights, limited access to and high costs of biomarkers restrict their routine use in practice (*McKhann et al., 2011*). This gap underscores the need for more efficient classification methods to facilitate early intervention because clinicians must manually inspect numerous MRI slices—a labor-intensive process prone to diagnostic error.

Recent advances in artificial intelligence (AI) have produced more accurate AD classification techniques. *Mora-Rubio et al. (2023)* introduced a DL pipeline that assigned structural MRI scans to five diagnostic categories: CN, EMCI, MCI, LMCI, and AD. The authors analyzed 2,559 T1-weighted images from the ADNI and the Open Access Series of Imaging Studies (OASIS). After FreeSurfer preprocessing and spatial data augmentation, the study compared EfficientNet, DenseNet, a custom Siamese CNN, and a Vision Transformer (ViT). The best-performing model achieved accuracies of 89% (AD *vs* CN), 80% (LMCI *vs* CN), 66% (MCI *vs* CN), and 67% (EMCI *vs* CN), demonstrating that DL approaches can effectively discriminate between healthy and diseased brains across multiple stages.

*Basaia et al. (2019)* developed a DL algorithm that predicts AD diagnosis and the likelihood of MCI conversion to AD from a single cross-sectional structural MRI scan. Their convolutional neural network distinguished AD, converting MCI (cMCI) and stable MCI (sMCI) with high accuracy, achieving its best performance in differentiating AD from healthy controls (HC): 99% using images from the ADNI database and 98% with a combined ADNI + Milan dataset. The model also distinguished cMCI from sMCI with up to 75% accuracy, showing no significant differences between ADNI and non-ADNI images. These findings highlight the potential of CNN-based methods to discriminate AD and MCI from HC and to predict conversion to AD within 36 months.

Given the limited availability of neuroimaging data, recent efforts have focused on models trained exclusively on clinical variables. *Yi et al. (2023)* integrated clinical data, neuropsychological scores, imaging-derived biomarkers, and genotypic information from the ADNI and NACC cohorts into an enhanced GB framework (XGBoost). The algorithm iteratively adds regression trees by minimizing the negative gradient of the loss function. Shapley Additive Explanations (SHAP) quantified the direction and magnitude of each feature's contribution, enhancing interpretability. The framework achieved high sensitivity (81.21%/74.85%), specificity (92.18%/89.86%), accuracy (87.57%/80.52%), the area under the receiver-operating characteristic curve (0.91/0.88), and positive/negative clinical utility indices (0.71/0.56 and 0.75/0.68) on the ADNI and NACC datasets, respectively.

*Rangegowda et al. (2023)* proposed a clinical-data framework for predicting progression from MCI to AD and for stage-wise classification. Using the ADNI dataset, they extracted demographic variables (age, education), disease progression rates, and cognitive test scores, partitioning the data into 70% for training and 30% for testing and reserving 10% of the training set for validation. The researchers evaluated multi-layer perceptron (MLP), random forest, support vector machine, and decision tree classifiers across binary and multiclass tasks involving AD, LMCI, EMCI, and CN. The MLP exhibited the best performance, achieving accuracies of 99.97% (AD *vs* LMCI), 99.57% (AD *vs* EMCI), 99.96% (AD *vs* CN),

and 95.05% (EMCI *vs* CN and LMCI *vs* CN). In multiclass scenarios, the model attained 99.97% (AD—LMCI—CN), 91.20% (EMCI—LMCI — AD), 86.25% (CN—EMCI—LMCI), 91.94% (CN—LMCI—AD), 85.14% (CN—EMCI—AD), and 77.50% when all four categories were jointly classified (AD—LMCI—EMCI—CN). These findings highlight the potential of MLP-based approaches for accurate AD staging using exclusively clinical data.

Recent investigations suggest that clinical variables offer several advantages over MRI for staging AD. While MRI scanners and biomarker assays remain inaccessible in many settings, healthcare providers can acquire clinical data at substantially lower cost in nearly all healthcare environments. Their broad availability and affordability enhance the feasibility of early detection and stratification across diverse populations. Consequently, this study designs DL and ML models to classify AD stages solely from clinical information.

# MATERIALS AND METHODS

## Database

This study utilized data from ADNI. Launched in 2003 as a public-private partnership, ADNI aims to determine whether MRI, positron emission tomography (PET), other biomarkers, and clinical assessments can reliably measure cognitive impairment and Alzheimer's disease progression, thereby promoting collaborative research. The database also includes comprehensive clinical data for all participants.

### Data processing and feature selection

The dataset was acquired in August 2023 and contains 13,205 patient records. It encompasses clinical variables, neuropsychological biomarkers derived from imaging, and APOE-4 status. In total, 113 features were available for modelling.

Following the feature-selection scheme proposed by *Yi et al. (2023)*, we first discarded variables with >50% missing values. We then evaluated the surviving candidates using three complementary selectors—Information Gain (filter), Boruta (wrapper), and Elastic Net (embedded)—thereby covering all three feature-selection paradigms. (i) Information Gain served as a fast entropy-based filter, removing variables with negligible relevance. (ii) Boruta, a random-forest wrapper, retained all relevant features and captured nonlinear interactions overlooked by univariate filters. (iii) An Elastic Net model ($\alpha = 0.5$) introduced joint L1/L2 regularisation, shrinking coefficients, and eliminating redundant, highly correlated predictors. Features endorsed by at least two of these methods were kept, yielding 27 clinically meaningful attributes (Table 1). The preprocessing step excluded patients missing selected features, resulting in a final dataset of 1,846 patients: 583 with CN, 122 with AD, 696 with EMCI, and 445 with LMCI. Figure 1 illustrates the class distribution. Stratified sampling preserved this distribution when the data were split into training (80%) and test (20%) sets.

## Class-balancing techniques

All class-balancing techniques were restricted to the training partition to preserve the integrity of the validation process. The original class distribution in the independent test

**Table 1 Selected features to train the model.** The first row indicates the feature type, the second row provides its abbreviation as found in the ADNI database, and the third row presents its meaning. The numbers accompanying the data refer to the position of each datum within the vector.

| Type of feature | Genetic | Demographic information | Neuroimaging-extracted biomarkers | Neuropsychological |
|---|---|---|---|---|
| **Name of feature** | | | | ADAS11 (Alzheimer's Disease Assessment Scale-Cognition 11 items) [5] |
| | | | | ADAS13 (Alzheimer's Disease Assessment Scale-Cognition 13 items) [6] |
| | | | | ADASQ4 (Score from Task 4 of the Alzheimer's Disease Assessment Scale) [7] |
| | | | Ventricles (Volume of ventricles) [20] | MMSE (Total Score of Mini-Mental State Examination) [8] |
| | | | Hippocampus (Volume of hippocampus) [21] | FAQ (Total Score of Functional Activities Questionnaire) [9] |
| | | AGE (Patient's age) [0] | WholeBrain (Volume of Whole Brain) [22] | MOCA (Total Score of Montreal Cognitive Assessment) [10] |
| | | PTMARRY (Patient's Marital stauts) [3] | Entorhinal (Volume of entorhinal) [23] | CDRSB (Clinical Dementia Rating-Sum of Boxes Score) [11] |
| | APOE4 (Number of APOE-ε44 alleles) [4] | PTGENDER (Patient's sex) [1] | Fusiform (Volume of fusiform) [24] | RAVLT_immediate (Rey's Auditory Verbal Learning Test_Immediate Recall) [12] |
| | | PTEDUCAT (Patient's time of education) [2] | MidTemp (Volume of middle temporal gyrus) [25] | RAVLT_learning (Rey's Auditory Verbal Learning Test_Learning) [13] |
| | | | ICV (Volume of intracranial) [26] | RAVLT_forgetting (Rey's Auditory Verbal Learning Test_Forgetting) [14] |
| | | | | RAVLT_perc_forgetting (Rey's Auditory Verbal Learning Test_Percent Forgetting) [15] |
| | | | | LDELTOTAL (Delayed total recall) [16] |
| | | | | TRABSCOR (Trail Making Test Part B Time) [17] |
| | | | | mPACCdigit (Modified Preclinical Alzheimer Cognitive Composite with Digit test) [18] |
| | | | | mPACCtrailsB (Modified Preclinical Alzheimer Cognitive Composite with Trails test) [19] |

sets was left intact, thereby preventing information leakage and ensuring that performance estimates reflect the model's ability to generalize to naturally imbalanced data.

### Synthetic minority over sampling technique

Synthetic minority oversampling technique generates synthetic samples for minority classes by interpolating between existing data points. It identifies the k-nearest neighbors of a sample within the minority class and creates new instances along the line segments connecting the original sample to its neighbors. This approach not only balances class distribution but also introduces variability that enhances model generalization (*Chawla et al., 2002*). As a preprocessing step, SMOTE significantly improved classification performance, particularly for minority classes, by reducing overfitting to the majority classes and enabling models to more effectively capture patterns associated with the early and late stages of cognitive decline.

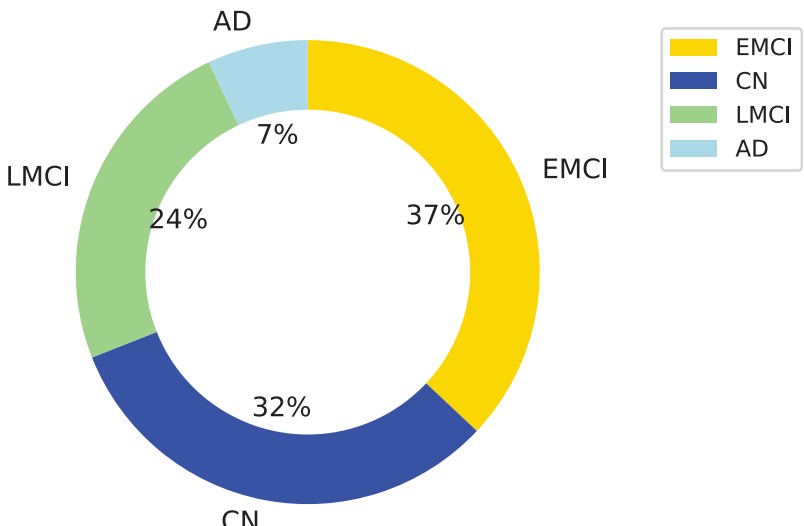

**Figure 1 Distribution of the dataset used in this study across four diagnostic categories: Alzheimer's disease (AD), cognitively normal (CN), early mild cognitive impairment (EMCI), and late mild cognitive impairment (LMCI).** The donut chart shows the percentage of samples per class, with LMCI representing the largest portion (24%), followed by CN (32%), EMCI (37%), and AD (7%). This class imbalance was considered in the model evaluation process.

### SMOTE-TOMEK

The SMOTE-Tomek method integrates two complementary techniques, SMOTE and Tomek Links elimination, to address class imbalance in datasets. This hybrid method both balances class distribution through synthetic sample generation and improves separability by removing ambiguous instances. It has proven particularly effective in classification tasks where decision boundaries are unclear and prone to overfitting. In the first stage, SMOTE generates synthetic samples for the minority class by interpolating feature values between selected minority instances and their nearest neighbors in the feature space. While SMOTE significantly improves class balance, placing synthetic points in regions where class overlap occurs may introduce noise near decision boundaries, potentially compromising model generalization. To address this issue, the second stage applies the Tomek Links technique. Tomek Links identifies pairs of instances from opposite classes that are each other's nearest neighbors, representing regions of class ambiguity. Removing these pairs refines the dataset by eliminating overlapping and conflicting examples, thereby reducing noise and enhancing the clarity of boundaries. As a result, the SMOTE-Tomek method improves data quality by mitigating noise introduced during oversampling and eliminating inherently problematic instances from the original dataset (*Batista, Bazzan & Monard, 2003*).

### Adaptive synthetic sampling

The adaptive synthetic sampling (ADASYN) algorithm addresses the challenge of class imbalance in ML by generating synthetic data points for the minority class, particularly targeting instances that are harder to classify (*He et al., 2008*). In imbalanced datasets, the

dominance of the majority class often leads to biased models and poor predictive performance for the minority class. ADASYN mitigates this issue by adaptively focusing on regions of the feature space where classification is most difficult for minority class samples. The algorithm operates through the following key steps:

- ADASYN identifies minority class instances that are more challenging to classify based on their proximity to majority class samples. Instances in more ambiguous regions receive more synthetic samples, while those in the well-separated areas receive fewer or none.
- It generates synthetic samples by interpolating between selected examples of the minority class and their nearest neighbors. The quantity of synthetic data produced for each instance is proportional to its classification difficulty.
- By adaptively generating samples, ADASYN improves classifier performance, reduces bias toward the majority class, and enhances generalization, particularly for underrepresented categories.

This strategy balances class distributions by generating synthetic samples tailored to reinforce decision boundaries, thereby enhancing the model's robustness and predictive accuracy.

### NearMiss

The NearMiss method is an under-sampling technique designed to address class imbalance by selecting a subset of majority class (negative) examples that are closest to the minority class (positive) instances, based on a predefined distance metric (*Mani & Zhang, 2003*). By focusing on these strategically chosen samples, NearMiss reduces class-imbalance skew and enhances the model's ability to learn the distinguishing features of the minority class. The method comprises three primary variations:

- **NearMiss-1:** Selects majority class samples with the smallest average distances to the three nearest minority class instances. This variation emphasizes boundary regions, capturing the most relevant negative examples for positive class decision margins.
- **NearMiss-2:** Selects majority class samples with the smallest average distances to the three farthest minority class instances. This approach ensures that the selected negatives are generally close to the overall distribution of the minority class.
- **NearMiss-3:** For each minority class instance, the algorithm selects a fixed number of the nearest majority class samples. This selection strategy ensures a balanced local distribution by surrounding each positive instance with an equal number of nearby negative examples.

## Model architectures
### Traditional ML models

- **Extra-Trees Classifier (ETC)**
  The ETC, or Extremely Randomized Trees, is a variant of the Random Forest algorithm that introduces a higher degree of randomness in the construction of individual decision

trees. ETC's additional randomness can enhance computational efficiency and increase robustness against overfitting in specific scenarios (*Géron, 2022*). The following section outlines the primary steps in building an ETC:

1. **Feature selection:** At each node, the algorithm randomly selects a subset of features $\mathscr{F} \subset \mathscr{X}$, where $\mathscr{X}$ denotes the full set of $d$ features in the dataset. While this step is similar to that used in Random Forests, ETC increases stochasticity by enforcing a fixed random subset.

2. **Random split point:** For each feature $f \in \mathscr{F}$, a split point $t$ is randomly selected from the range of values observed for $f$ in the current node. ETC differs from Random Forests, which determine the split point by optimizing a criterion to minimize impurity.

3. **Split criterion:** Each candidate pair $(f, t)$ is evaluated using a cost function, such as entropy or the Gini index. Selecting the combination that minimizes node impurity yields the optimal split.

   Formally, the optimal splitting point at a node $N$ is given by:

   $$\text{Optimal} = \arg \min_{f \in \mathscr{F}, t \in \mathscr{T}} \text{Impurity}(N, f, t)$$

   where $\mathscr{T}$ is the set of possible split points for feature $f$, and the impurity $\text{Impurity}(N, f, t)$ is commonly calculated using the Gini index:

   $$\text{Gini index} = 1 - \sum_{k=1}^{K} p_k^2.$$

   Here, $K$ represents the number of classes, and $p_k$ denotes the proportion of samples belonging to class $k$ at node $N$.

- **Random Forest (RF)**

  The RF is an ensemble learning method that constructs multiple decision trees, each trained on a different subset of the data and features (*Breiman, 2001*):

  1. **Bootstrap aggregating (Bagging):** Multiple bootstrap samples $\mathscr{B}_i$ are drawn with replacement from the training set $\mathscr{D}$, and each sample is used to train an individual decision tree $T_i$.

  2. **Random feature selection:** At each node of a decision tree, a random subset of features $\mathscr{F} \subset \mathscr{X}$ is selected from the total set of $d$ available features.

  3. **Node splitting:** From the subset $\mathscr{F}$, the algorithm selects the optimal feature $f$ and split point $t$ by minimizing an impurity measure, such as entropy or the Gini index.

The final prediction of the Random Forest is obtained by aggregating the predictions of all individual trees using majority voting for classification tasks or averaging for regression:

$$\hat{y} = \frac{1}{n_{\text{trees}}} \sum_{i=1}^{n_{\text{trees}}} T_i(\mathbf{x})$$

where $T_i(\mathbf{x})$ denotes the prediction of the $i$-th tree for the input instance $\mathbf{x}$.

- **Support vector classifier**

  Support vector classifier (SVC) is a widely used class of ML algorithms designed to solve both classification and regression tasks involving linear (flat) and nonlinear (curved) decision boundaries. The primary objective of SVM is to maximize the margin between classes, optimizing the separation as a broad "highway" rather than a narrow boundary (*Géron, 2022*).

  The decision function linearly combines the features of a data point using weights $\omega$ and a bias $\omega_0$. The result is passed through a sign function to determine the predicted class:

  $$f(\omega; x) = \text{sign}(\omega^T x + \omega_0).$$

  A positive result assigns the instance to class $+1$; a negative result assigns it to class $-1$. SVM employs the hinge loss function to penalize misclassified points and those within the margin boundaries. Hinge loss is given by:

  $$\max(0, 1 - y_{\text{true}}(\omega^T x + \omega_0))$$

  where:

  - If $y_{\text{true}} = +1$ and $\omega^T x + \omega_0 < 0$, misclassifying the sample incurs a high penalty.
  - If $y_{\text{true}} = +1$ and $0 < \omega^T x + \omega_0 < 1$, the classification is correct but within the margin, leading to a lower penalty.
  - If $y_{\text{true}} = +1$ and $\omega^T x + \omega_0 > 1$, no penalty is applied.

    The same logic applies for $y_{\text{true}} = -1$.

    A regularization term is included in the objective function to encourage a wider margin. This regularization term governs the model's complexity and is weighted by a hyperparameter $\lambda$. The following expression defines the complete loss function:

    $$L(\omega) = \frac{1}{m} \sum_{i=1}^{m} \max(0, 1 - y_{\text{true}}^i(\omega^T x^i + \omega_0)) + \lambda ||\omega||_2^2$$

    where $m$ is the number of training examples and $\lambda$ regulates the trade-off between margin maximization and classification error.

    The optimization objective involves determining the values of $\omega$ that minimize the loss function. This minimization results in a convex quadratic programming problem, thereby ensuring the existence of a unique global minimum. The problem can be reformulated in its dual form, particularly beneficial in high-dimensional feature spaces. The dual optimization problem is given by:

    $$\max_{\alpha} \sum_{j=1}^{m} \alpha_j - \frac{1}{2} \sum_{j=1}^{m} \sum_{k=1}^{m} \alpha_j \alpha_k y_{\text{true}}^j y_{\text{true}}^k (x^j)^T x^k$$

    subject to the constraints $\alpha_j \geq 0$ and $\sum_{j=1}^{m} \alpha_j y_{\text{true}}^j = 0$.

    Once the optimal values $\alpha_j$ are obtained, the weight vector $\omega$ can be recovered using:

    $$\omega = \sum_{j=1}^{m} \alpha_j y_{\text{true}}^j x^j.$$

With $\omega$ computed, the trained decision function classifies new data points:

$$f(x_{\text{new}}) = \text{sign}(\omega^T x_{\text{new}} + \omega_0) = \text{sign}\left(\sum_{j=1}^{m} \alpha_j y_{\text{true}}^j (x^j)^T x_{\text{new}} + \omega_0\right).$$

- **Gradient boosting**

  GB is an ensemble technique that constructs a strong predictive model by sequentially combining multiple weak learners (*Friedman, 2002*). The core idea is to build models iteratively, correcting the residual errors of the previous models at each step. The model achieves this by minimizing a specified loss function using a gradient descent approach (*Géron, 2022*). The mathematical formulation is:

  1. **Initial model:** The process begins with a constant base model $F_0(x)$, typically defined as the value that minimizes the loss function over the training data—often the mean of the target variable. The following presents the mathematical formulation:

  $$F_0(x) = \arg\min_{\gamma} \sum_{i=1}^{n} L(y_i, \gamma)$$

  where $L(y, F(x))$ is the loss function measuring the discrepancy between the true value $y$ and the model prediction $F(x)$.

  2. **Residual computation:** At each iteration $m$, the algorithm computes the pseudo-residuals, which correspond to the negative gradients of the loss function concerning the current predictions $F_{m-1}(x)$:

  $$r_{im} = -\left[\frac{\partial L(y_i, F(x_i))}{\partial F(x_i)}\right]_{F(x)=F_{m-1}(x)}.$$

  These residuals indicate how the model should adjust to minimize the prediction error.

  3. **Fitting the new model:** A new weak learner $h_m(x)$ is fitted to the residuals using a least squares approach:

  $$h_m(x) = \arg\min_{h} \sum_{i=1}^{n} [r_{im} - h(x_i)]^2.$$

  Aligning the model $h_m(x)$ with the structure of the current residual errors

  4. **Model update:** The ensemble model is updated by adding the newly fitted model $h_m(x)$, scaled by a learning rate $v$, to the previous model:

  $$F_m(x) = F_{m-1}(x) + vh_m(x).$$

  The learning rate $v \in (0, 1]$ regulates the contribution of each new model, providing a trade-off between training speed and performance.

  5. **Final prediction:** After $M$ iterations, the final prediction function is defined as:

  $$F_M(x) = F_0(x) + \sum_{m=1}^{M} vh_m(x).$$

This cumulative model integrates all weak learners constructed during training to form a strong predictor.

### DL models

- **CNNs**

  CNNs extract meaningful features from input data, forming the foundation of their functionality. Researchers have developed numerous CNN architectures in response to advances in computational power and DL; however, most architectures share a standard structure consisting of three main components: a feature extraction layer, a fully connected layer, and an output layer.

  The feature extraction layer consists of multiple convolutional kernels that map various features from the input data. Kernels learn and capture low-level to high-level patterns, followed by the application of an activation function. The fully connected layer connects each neuron from the previous layer to every neuron in the current layer, transforming spatial features into semantic representations. The output layer, commonly used for classification tasks, typically includes an optimizer and a softmax activation function to produce probabilistic outputs (*Gu et al., 2018*).

  $$M^l = \text{pool}\left( f\left( \text{norm}\left( \sum_{i=1}^{n} \left( M_i^{l-1} * K_i^l \right) + b^l \right) \right) \right). \tag{1}$$

  In this equation, $M^l$ denotes the output feature map at layer $l$, $M_i^{l-1}$ is the input feature map from the previous layer, $K_i^l$ represents the convolutional kernel, $b^l$ is the bias term, and $*$ indicates the convolution operation.

  Figure 2 illustrates the feature-extraction block. The network first applies three convolutional layers with 8, 16, and 32 filters, each employing the "selu" activation function (scaled exponential linear unit, SELU). It then applies a max pooling layer and a dropout layer with a rate of 0.5 to mitigate overfitting. Next, the network applies additional convolutional layers with 64, 128, and 256 filters, each followed by a max pooling layer and dropout to further refine the extracted features.

  Figure 3 presents the fully connected architecture. The network includes Fully connected layers with 1,024 to 16 neurons in descending order, respectively. The final output layer adapts according to the number of target classes. Each layer utilizes the "selu" activation function and incorporates batch normalization to stabilize training and accelerate convergence.

- **DigitCapsule-Net**

  The Capsule Network (Capsule-Net) architecture significantly advances modeling spatial and hierarchical relationships within data. Unlike traditional CNNs, which aggregate features *via* convolution and pooling operations, Capsule-Net employs convolutional blocks to learn features and then routes these features through capsules. This approach enhances classification accuracy by emphasizing critical activations and maintaining a richer input representation. Capsule-Net demonstrates robustness to

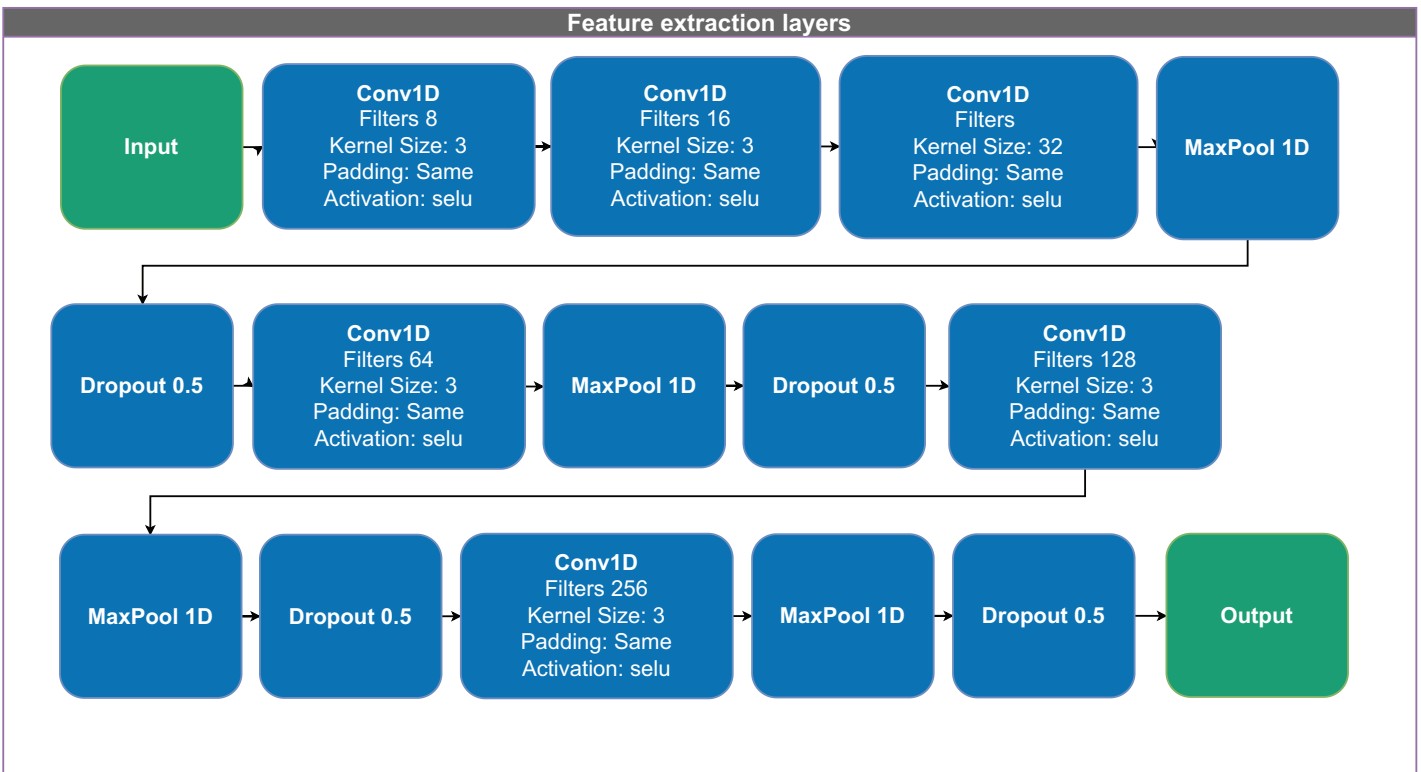

**Figure 2** **Architecture of the 1D convolutional neural network (CNN) used for feature extraction and classification.** The model receives a 1D input vector and applies a sequence of Conv1D layers with increasing filter sizes (8 to 256), kernel size of 3, and scaled exponential linear unit (SELU) activation. Dropout layers with a rate of 0.5 and MaxPooling1D layers are interleaved to reduce overfitting and dimensionality. The final representation is passed to the output layer for classification. All convolutional layers use same padding to preserve input dimensions across layers.

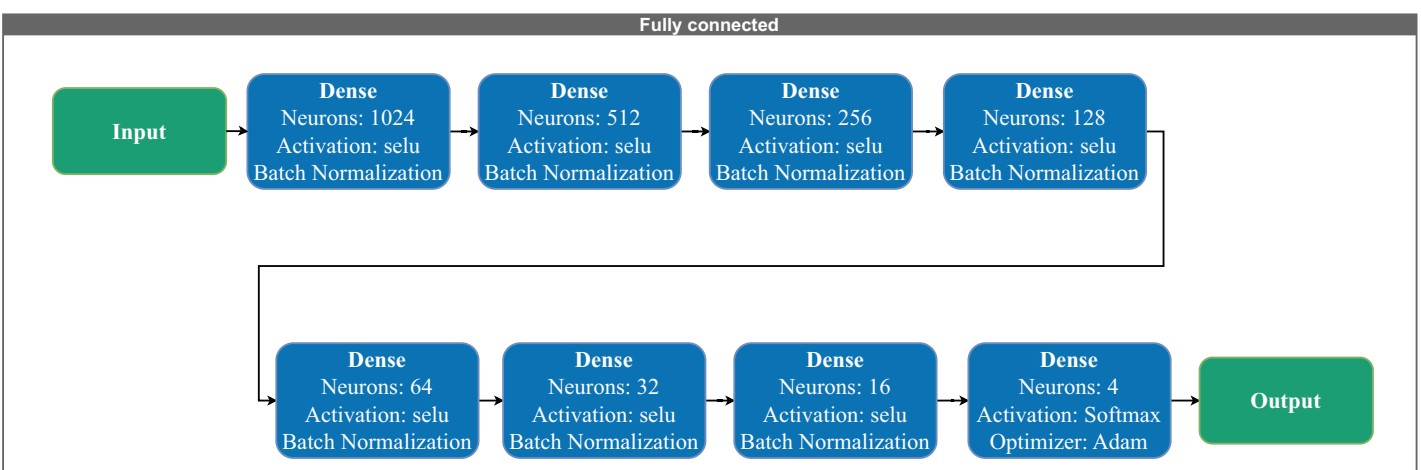

**Figure 3** **Fully connected layers consisting of fully connected layers with 1,024 to 16 neurons in descending order.** Each layer includes hyperparameters such as activation functions and normalization.

minor variations in input data, leading to more accurate and reliable image interpretations.

The DigitCapsule-Net layer is a core component of the Capsule-Net architecture and follows the primary (or central) capsules. This layer contains multiple capsules, each functioning as a small neural network that generates an activation vector. Each capsule actively represents a specific class, such as digits in image recognition tasks, and produces a vector that captures detailed information about the input. This vector-based representation enables the model to express and interpret the data more comprehensively than traditional scalar outputs (*Sabour, Frosst & Hinton, 2017*).

A key innovation of the DigitCapsule-Net is its use of dynamic routing, a process that determines the strength of the connections between the outputs of primary capsules and the higher-level capsules in the DigitCapsule-Net layer. During dynamic routing, the predictions from the lower-level capsules are iteratively refined based on their agreement with the outputs of the higher-level capsules. This mechanism allows the network to preserve more information than max-pooling, resulting in more precise data representations.

DigitCapsule-Net derives class probabilities from the length of each output vector, which reflects the likelihood of a particular class being present. Additionally, the orientation of these vectors encodes detailed properties of the represented objects, such as their pose, size, and deformation. This dual encoding enhances the interpretability and accuracy of classification tasks, offering a more nuanced understanding of the input data. The DigitCapsule-Net exhibits strong robustness to variations in object orientation, illumination, and position, making it particularly suitable for complex recognition scenarios (*Sabour, Frosst & Hinton, 2017*; *Dombetzki, 2018*; *Holguin-Garcia et al., 2024*). In the proposed model, the DigitCapsule-Net layer is integrated following the convolutional and pooling layers, as shown in Fig. 4. The dimensions are adjusted to facilitate the efficient processing of capsule vectors. The model first normalizes and compresses the output values using an activation function to ensure they remain within an appropriate range. The model then passes this compressed output to DigitCapsule-Net, which applies linear transformations and produces a multidimensional tensor representing the learned class-specific features.

- **Transformer encoder**

The Transformer encoder captures and models contextual information by establishing relationships among input vectors. Unlike recurrent language models such as long short-term memory (LSTM) networks, it employs self-attention mechanisms that effectively capture the contextual representation of features within a sequence of vectors. Each encoder layer includes multiple attention heads, feed-forward layers, and positional encoding, which retains information about the order of the input elements (*Yan et al., 2019*).

Multiple attention heads apply self-attention in parallel; each head partitions the input into a query (Q), key (K), and value (V) vectors, allowing the model to process different representation subspaces simultaneously. The outputs of all attention heads are

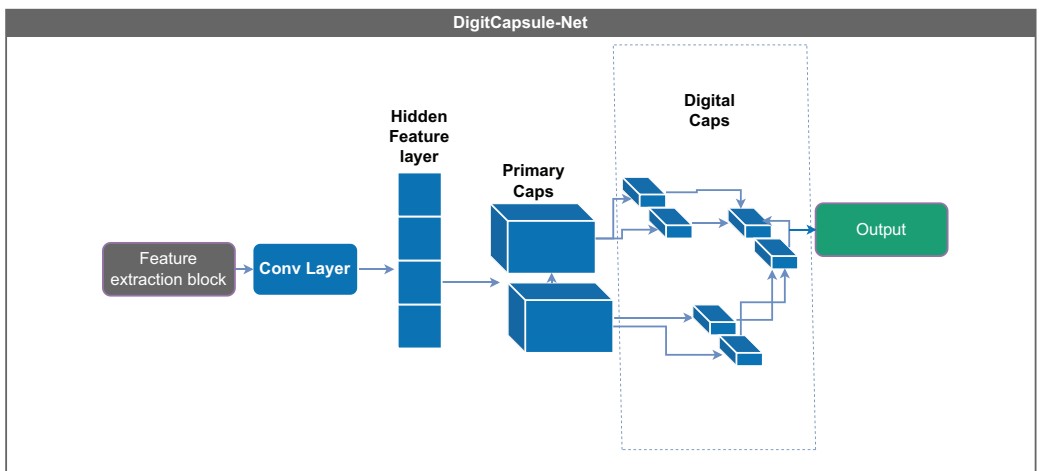

**Figure 4 DigitCapsule-Net model.** Processes information using dynamic routing (*Sabour, Frosst & Hinton, 2017*).

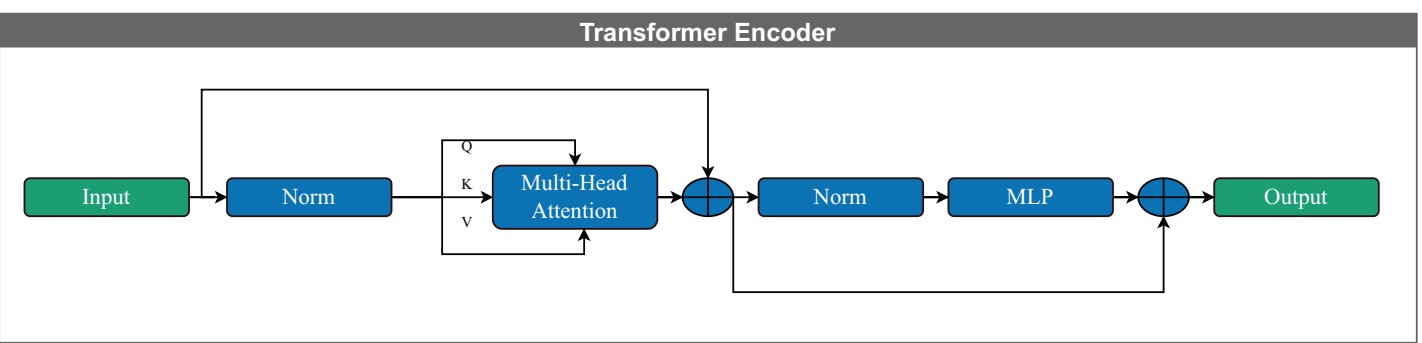

**Figure 5 Schematic representation of a Transformer encoder block. The input is first normalized and then processed through a multi-head self-attention mechanism, which computes contextualized representations using query (Q), key (K), and value (V) vectors.** The output of the attention layer is added to the input *via* a residual connection and followed by layer normalization. This is passed through a feed-forward neural network (MLP), again followed by residual addition and normalization, yielding the final encoder output. This architecture enables efficient modeling of long-range dependencies in sequential data.

concatenated and passed through a learned linear transformation to form the final output, a process known as *multi-head attention* (*Vaswani et al., 2017*; *Bravo-Ortíz et al., 2024c, 2024a*).

Figure 5 illustrates the Transformer encoder architecture used in this study. The following equation defines the self-attention mechanism:

$$\text{Attn}(\mathbf{Q}, \mathbf{K}, \mathbf{V}) = \text{softmax}\left(\frac{\mathbf{Q}\mathbf{K}^T}{\sqrt{d_k}}\right)\mathbf{V}. \tag{2}$$

In this equation, Q represents the query vector corresponding to the current token, K denotes the key vector, and V is the value vector containing relevant contextual information. The term $\sqrt{d_k}$ serves as a normalization factor for scaling the dot product, where $d_k$ is the dimensionality of the key vectors.

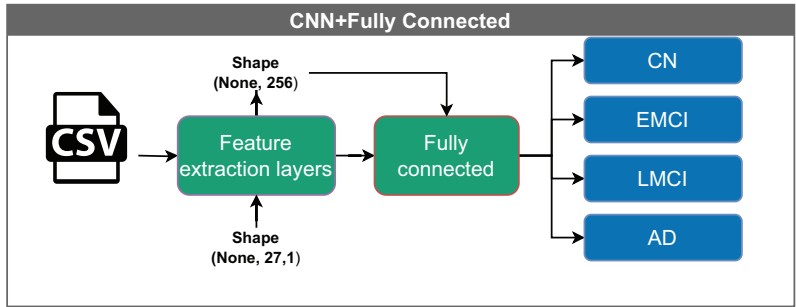

**Figure 6 Architecture of the CNN model with a fully connected layer.** The model receives input feature vectors of shape (None, 27, 1), which are processed by feature extraction layers, resulting in a representation of shape (None, 256). These vectors are then passed through a fully connected layer, reducing the dimensionality to (None, 16), before being classified into one of four categories: CN, EMCI, LMCI, and AD. The "None" dimension denotes the variable input batch size.

## Configuration of DL models

### Convolutional neural network–fully connected

The CNN with a fully connected layer (Fig. 6) represents the most basic configuration and comprises two main components: a feature extraction block and a classification block implemented *via* a fully connected layer at the output.

The input to the model is a vector with dimensions (None, 27, 1). The convolutional filters process the data into a shape of (None, 256), which is then reduced to (None, 16) before it enters the classification layer. The term "None" denotes the variable batch size.

### Convolutional neural network–DigitCapsule-Net

Convolutional neural network–DigitCapsule-Net (CNN+DigitCapsule-Net), illustrated in Fig. 7, receives input vectors with dimensions (None, 27, 1). These inputs are first processed through a feature extraction layer, resulting in a tensor of shape (None, 1, 256). The model passes the output to the DigitCapsule-Net, which produces a final representation of shape (None, 3, 16) for classification. As before, "None" indicates the variable input batch size.

### Convolutional neural network–Transformer encoder

The convolutional neural network–transformer encoder (CNN+TF) shown in Fig. 8, takes input vectors of shape (None, 27, 1). These vectors are passed through feature extraction layers, producing an output of shape (None, 1, 256). Within the Transformer block, the data undergo processing through multiple attention heads while maintaining the same dimensionality. A multilayer perceptron performs the classification in the final stage.

### Convolutional neural network–Transformer encoder + DigitCapsule-Net (CNN +TF+DigitCapsule-Net)

The hybrid model illustrated in Fig. 9 integrates several advanced components for data processing. Input vectors of shape (None, 27, 1) are first passed through a shared feature extraction block, resulting in an intermediate representation of shape (None, 1, 256). This

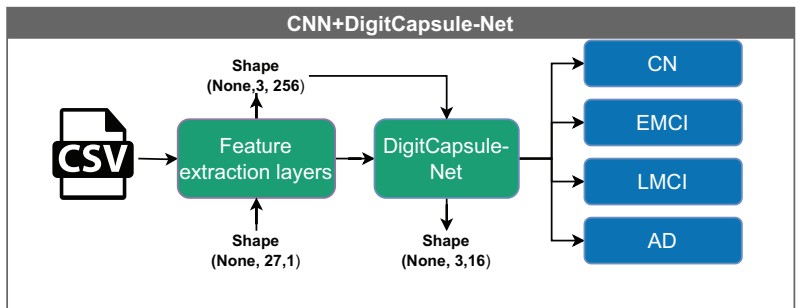

**Figure 7 Architecture of the CNN model with a DigitCapsule-Net.** The model receives input vectors of shape (None, 27, 1), which are processed by feature extraction layers to yield an output of shape (None, 1, 256). The tensor is then passed to the DigitCapsule-Net Block, producing a final representation of shape (None, 3, 16) for classification into one of four categories: CN, EMCI, LMCI, and AD. The "None" dimension denotes the variable input batch size.

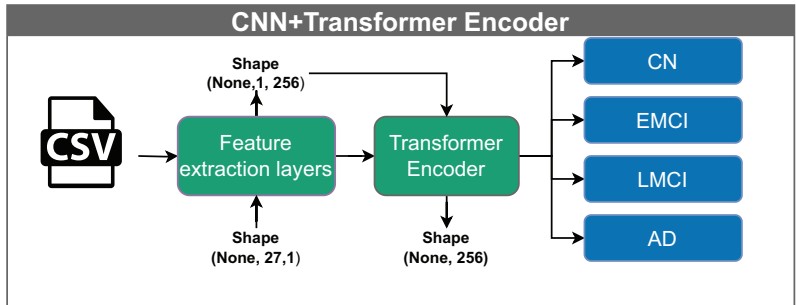

**Figure 8 Architecture of the CNN model with a Transformer encoder.** The model receives input vectors of shape (None, 27, 1), which are transformed by feature extraction layers into tensors of shape (None, 1, 256). This output is processed through multiple attention heads within the Transformer encoder, preserving the shape, before being classified into one of four categories: CN, EMCI, LMCI, and AD. The "None" dimension denotes the variable input batch size.

output passes through a Transformer encoder, which captures complex feature relationships *via* attention mechanisms without altering dimensionality. Finally, the DigitCapsule-Net receives the output and produces a final representation of shape (None, 3, 16) for classification. This architecture combines the efficiency of CNNs, the contextual modeling capabilities of Transformers, and the hierarchical feature representation of capsule networks, thereby leveraging the strengths of each.

## Evaluation metrics and validation strategy
### *Metrics*
By emphasizing the differences between false positives (FP), false negatives (FN), true positives (TP), and true negatives (TN), *Tabares-Soto et al. (2021)* illustrate the importance of metrics in the evaluation of a model. The following are the most crucial metrics:

- **Accuracy**
  Accuracy evaluates a classification model's predictive performance. To compute it, one divides the total number of correct predictions by the total number of predictions.

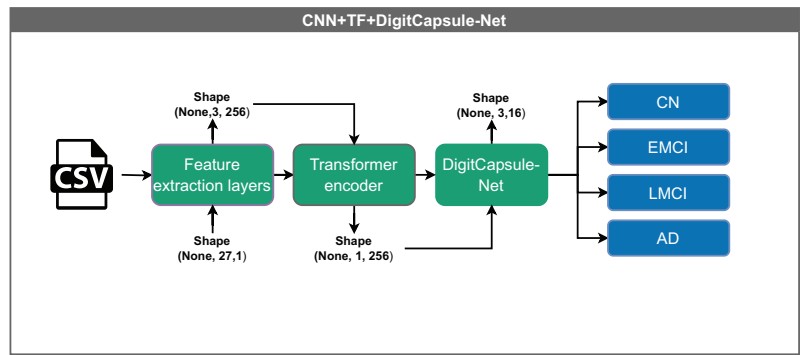

**Figure 9 Architecture of the hybrid model combining CNN, Transformer encoder, and DigitCapsule-Net.** The model receives input vectors of shape (None, 27, 1), which are processed by shared feature extraction layers to generate a representation of shape (None, 1, 256). The intermediate output flows through a Transformer encoder, which captures contextual dependencies without altering its shape. Subsequently, the output is processed by the DigitCapsule-Net, producing a final representation of shape (None, 3, 16) for classification into one of four categories: CN, EMCI, LMCI, and AD. The "None" dimension denotes the variable input batch size.

Accuracy is a value ranging from 0 to 1, representing the percentage of accurate predictions.

$$Accuracy = \frac{TP + TN}{TP + TN + FP + FN}. \tag{3}$$

- **Precision**
  The precision metric gauges the ratio of correctly identified positive cases by a model to the total cases identified as positive, encompassing both true and false positives.

$$Precision = \frac{TP}{TP + FP}. \tag{4}$$

- **Recall**
  Also known as sensitivity, it shows the ability of the classifier to display correct predictions.

$$Recall = \frac{TP}{TP + FN}. \tag{5}$$

- **F1**
  F1 is a metric used to assess a model's ability to precisely identify positive and negative cases, particularly in scenarios where the data is imbalanced, and the positive class is infrequent. It is determined as the harmonic mean of precision and recall, proving especially valuable in situations with uneven class distribution.

$$F1 = 2 * \frac{Precision * Recall}{Precision + Recall}. \tag{6}$$

- **Confusion matrix**
  The confusion matrix is a tabular representation that encapsulates the correspondence between a model's predictions and the actual labels of the data. It incorporates true positives (TP), true negatives (TN), false positives (FP), and false negatives (FN). This

matrix proves valuable for visually summarizing a model's performance, highlighting both its successes and errors. Each row in the matrix corresponds to the actual class, and each column indicates the number of predictions made for each class. Additionally, it helps identify instances where the model misclassifies one class as another.

### Cross-validation

Cross-validation (CV) is a resampling technique used to evaluate a model's performance by partitioning the dataset into $k$ mutually exclusive subsets (folds) of approximately equal size. At each iteration, one fold serves as the validation set, while the remaining $k - 1$ folds constitute the training set. The process continues until all folds have individually served as the validation set. Averaging the $k$ resulting performance scores provides a more reliable estimate of the model's generalization ability. To preserve the original class distribution within every fold and avoid biased estimates in the presence of class imbalance, we adopt stratified 10-fold cross-validation, as illustrated in Eq. (7).

$$\hat{\mathcal{M}} = \frac{1}{k} \sum_{i=1}^{k} \mathcal{M}_i, \tag{7}$$

where $\mathcal{M}_i$ denotes the performance metric computed on the $i$-th validation fold.

### Statistical comparison of model performance

This study rigorously evaluated model performance differences using several non-parametric statistical tests, as recommended by *Rainio, Teuho & Klén (2024)*. These methods are particularly appropriate in scenarios where the assumptions underlying parametric tests—such as normality and homoscedasticity—are violated, which is often the case with cross-validated performance metrics.

- **Friedman test**
  The Friedman test serves as a non-parametric alternative to repeated-measures ANOVA, designed to detect differences in treatments across multiple testing conditions. Within the context of ML, it assesses whether statistically significant performance differences exist among various models evaluated on the same datasets. The test ranks model performance scores per dataset and analyzes these ranks to determine the significance of observed differences. Its robustness to violations of normality makes it particularly suitable for comparing more than two models simultaneously (*Rainio, Teuho & Klén, 2024*).

- **Wilcoxon signed-rank test**
  This study applied the Wilcoxon signed-rank test for pairwise model comparisons. This non-parametric method evaluates whether a significant difference exists between two related samples, such as the performance scores of two models across the same cross-validation folds. By accounting for both the magnitude and direction of the differences, the test provides a more comprehensive assessment than the sign test. It is especially well-suited for situations in which the assumption of normality is not upheld (*Rainio, Teuho & Klén, 2024*).

- **Levene's test for equality of variances**

  Analyzing performance variability is essential, as models with similar mean accuracies may differ substantially in consistency. Levene's test was employed to assess the homogeneity of variances across model performance metrics. Unlike other variance equality tests, Levene's method is less sensitive to deviations from normality, making it an appropriate choice for this analysis (*Rainio, Teuho & Klén, 2024*).

## Hyperparameter tuning

### ML hyperparameter tuning with grid search and pipelines

This study employed a combination of *grid search* and *pipeline integration* to enhance the performance of traditional ML models. Grid search is a widely used methodology that systematically explores a predefined subset of the hyperparameter space to identify the optimal configuration. Although effective, grid search can be computationally expensive due to the exponential growth in combinations as the number of hyperparameters increases. Consequently, practitioners often restrict the search to carefully selected regions of the parameter space, typically assuming mutual independence among the hyperparameters (*Liashchynskyi & Liashchynskyi, 2019*).

We ensured reproducibility and efficiency by embedding all hyperparameter tuning procedures within a pipeline architecture, consistent with the implementation described in *Pedregosa et al. (2011)*. Pipelines facilitate the seamless integration of preprocessing steps (feature selection, scaling), model training, and evaluation. This approach minimizes the risk of data leakage and promotes methodological consistency across experiments.

Each model underwent independent optimization for both balanced and imbalanced datasets. The optimized hyperparameters for the three-class classification task are presented in Table 2.

### DL model tuning

The hyperparameters of DL models were carefully tuned to ensure optimal configuration during the training process. The categorical cross-entropy loss function was employed in conjunction with the Adam optimizer, configured with a learning rate of 0.001. All models were trained with a batch size of 256 for a total of 500 epochs, incorporating early stopping to prevent overfitting.

The following section provides a detailed description of the hyperparameters used.

- **Batch normalization (BN)**

  The features in each data map are normalized using the BN to have a mean of 0 and a variance of 1, enabling rescaling and retranslating the distribution. A faster rate of learning is made possible by this training method (*Ioffe & Szegedy, 2015*). The following equation denotes it:

$$BN(x, \gamma, \beta) = \beta + \gamma \frac{x - \mathrm{E}[X]}{\sqrt{Var[X] + \varepsilon}} \qquad (8)$$

**Table 2 Hyperparameter optimization for three class divided into each of the models, which in turn is divided into balanced data using SMOTE and imbalanced data.**

| Models | Hyperparameter | CN-EMCI-LMCI | | AD-EMCI-LMCI | | AD-CN-LMCI | | AD-CN-EMCI | |
|---|---|---|---|---|---|---|---|---|---|
| | | Imbalanced | Balanced | Imbalanced | Balanced | Imbalanced | Balanced | Imbalanced | Balanced |
| ETC | min_samples_split | 2 | 2 | 2 | 2 | 2 | 2 | 2 | 2 |
| | n_estimators | 100 | 200 | 150 | 300 | 150 | 150 | 250 | 150 |
| | random_state | 20 | 50 | 40 | 40 | 20 | 10 | 30 | 50 |
| SVM | C | 10 | 10 | 10 | 10 | 10 | 10 | 10 | 10 |
| | gamma | auto | scale | auto | auto | auto | scale | scale | scale |
| | Kernel | rbf | rbf | rbf | rbf | rbf | rbf | rbf | rbf |
| RFC | min_samples_split | 2 | 4 | 2 | 2 | 4 | 2 | 4 | 4 |
| | n_estimators | 250 | 300 | 100 | 300 | 50 | 400 | 50 | 300 |
| | random_state | 20 | 30 | 20 | 50 | 10 | 10 | 10 | 20 |
| GB | learning_rate | 0.1 | 0.1 | 0.1 | 0.1 | 0.05 | 0.1 | 0.1 | 0.05 |
| | max_depth | 7 | 7 | 3 | 7 | 7 | 3 | 5 | 7 |
| | n_estimators | 200 | 200 | 200 | 200 | 200 | 200 | 200 | 200 |
| | random_state | 10 | 50 | 50 | 50 | 20 | 10 | 40 | 20 |

where:

$\gamma$ = The re-scaling scalar.

$\beta$ = re-translation scalar.

$E[X]$ = expectation.

$Var[X]$ = variance.

- **Scaled exponential linear unit**

  This nonlinear activation function, proposed by *Klambauer et al. (2017)*, behaves linearly for positive input values and exhibits exponential behavior for negative ones. The function incorporates two constants: $\lambda$, approximately equal to 1.0507, and $\alpha$, the negative slope coefficient, with an approximate value of 1.67326. These parameters enable effective scaling and propagation of signals across multiple neural network layers. Furthermore, researchers consider the function self-normalizing because it maintains a constant mean and variance during forward propagation, thereby enhancing model stability (*Rasamoelina, Adjailia & Sinčák, 2020*).

  The following equation gives it:

$$\text{SELU}(x) = \lambda \begin{cases} x & \text{if } x > 0 \\ \alpha \cdot (\exp(x) - 1) & \text{if } x \leq 0 \end{cases}. \tag{9}$$

- **Dropout**

  The regularization technique known as *Dropout* is employed to prevent overfitting in neural networks. This method randomly drops out selected nodes and their connections during training, reducing the risk of co-adaptation among neurons and preventing the network from relying excessively on specific pathways. As a result, the model is less likely to memorize the training data and more capable of generalizing to unseen samples.

Beyond mitigating overfitting, Dropout contributes to the development of more robust networks by forcing them to function under varying structural configurations during training. This variability promotes model resilience and improves generalization. Additionally, Dropout facilitates the averaging of predictions and helps reduce variance at test time (*Srivastava et al., 2014*). Dropout is incorporated to prevent overfitting in this study, as illustrated in Fig. 2.

## Interpretability techniques

### Feature importance

The importance of features is a fundamental technique in interpreting ML models. It helps understand the model's behavior and identify biases and critical features. This method is crucial as artificial intelligence models have become increasingly complex and challenging to interpret due to scientific advancements (*Adler & Painsky, 2022*).

### Gradient-weighted class activation mapping

Gradient-weighted class activation mapping (Grad-CAM) is a widely adopted technique for interpreting CNNs by visualizing input regions that most influence the model's predictions for making predictions based on the information provided by the gradient. It computes the gradient of the predicted class score concerning the feature maps of the last convolutional layer, averages these gradients to derive weights, and multiplies them by the corresponding feature maps to produce a saliency map that highlights the essential input regions (*Selvaraju et al., 2016*).

### Shapley additive explanations values

Shapley additive explanations (SHAP) values constitute a widely adopted technique for interpreting the output of ML models. They provide insight into how individual input features contribute to a specific prediction. SHAP values is particularly valued for its consistency and fairness, as it is grounded in principles from cooperative game theory (*Marcílio & Eler, 2020*; *Meng et al., 2020*).

The theoretical foundation of SHAP lies in Shapley values, a concept from cooperative game theory that defines a fair method for distributing the payoff of a task among its participants. In the context of ML, the "participants" are the input features, and the "payoff" corresponds to the model's predicted output. SHAP values quantify each feature's contribution to the prediction by considering all possible combinations of features (*Marcílio & Eler, 2020*; *Meng et al., 2020*).

Formally, SHAP values assigns a value to feature $x_i$ by averaging its marginal contributions across all possible feature subsets.

Given a set of input features $\{x_1, x_2, \ldots, x_n\}$, the SHAP value $\phi_i$ is computed as:

$$\phi_i = \sum_{S \subseteq N \setminus \{i\}} \frac{|S|!(|N| - |S| - 1)!}{|N|!} (f(S \cup \{i\}) - f(S))$$

**Table 3 Results from ablation study of individual model components.** Each model was trained and evaluated independently on the same three-class classification task (AD *vs* EMCI *vs* CN). Metrics are averaged over cross-validation with standard deviation in parentheses.

| Architecture | Model | Precision [%] | Recall [%] | F1-score [%] | Accuracy [%] |
|---|---|---|---|---|---|
| Single Block | CNN | 86.48 | 86.25 | 86.23 | 86.25 |
| | TF | 23.18 | 48.04 | 31.24 | 48.04 |
| | DigitCapsule-Net | 88.03 | 87.86 | 87.81 | 87.86 |
| Combined | CNN + DigitCapsule-Net | 90.97 | 92.91 | 91.93 | 90.58 |
| | CNN + TF | 86.07 | 87.78 | 88.89 | 86.16 |

where:

- $\phi_i$ is the SHAP value for feature $x_i$.
- $N$ denotes the full set of input features.
- $S$ is any subset of features that does not include $x_i$.
- $f(S)$ is the model's prediction using only the features in subset $S$.
- $f(S \cup \{i\})$ is the model's prediction when feature $x_i$ is added to subset $S$.
- $\frac{|S|!(|N|-|S|-1)!}{|N|!}$ is a weighting factor that ensures a fair distribution based on all possible feature coalitions involving $x_i$.

## Hardware and computational resources

Google Colab was used for all experiments; in this case, using NVIDIA GP100GL (T4 PCIe 15 GB) with 250 W, CUDA Version 10.1, and RAM with 12 GB.

## RESULTS

### DL ablation

To assess the individual contribution of each architectural component—CNN, DigitCapsule-Net, and TF—to the final model performance, we conducted an ablation study. Each model was evaluated independently using the same data partition and evaluation protocol as the proposed architecture (CNN + DigitCapsule-Net).

Table 3 summarizes the performance of each isolated module. The CNN-only model achieved a precision of 86.48%, recall of 86.25%, and F1-score of 86.23%, while the DigitCapsule-Net-only model slightly outperformed CNN with an F1-score of 87.81%. In contrast, the Transformer-only model significantly underperformed, with a precision of 23.18% and an F1-score of 31.24%, which can be attributed to the relatively small dataset size and the lack of positional bias modeling in isolation.

These results confirm that each component brings valuable features to the overall architecture. The performance gains obtained by combining CNN and DigitCapsule-Net (91.93% F1-score) or adding TF (in another experiment) suggest that the fusion of local spatial features, hierarchical routing, and attention mechanisms is critical for robust classification across AD, EMCI, and CN groups.

**Table 4 Results obtained from all models evaluated in two classes include precision, recall, F1-score, and the accuracy obtained in 10 k-fold cross-validation.** Additionally, each result is presented for both imbalanced and balanced data using SMOTE method.

| Classes | Models | Precision [%] | | Recall [%] | | F1-score [%] | | Cross-validation [%] | |
|---|---|---|---|---|---|---|---|---|---|
| | | Imbalanced | Balanced | Imbalanced | Balanced | Imbalanced | Balanced | Imbalanced | Balanced |
| CN *vs* EMCI | CNN | 87.90 | 84.11 | 93.16 | 92.03 | 90.46 | 87.89 | 88.17 ± 3.17 | 90.01 ± 2.83 |
| | CNN + DigitCapsule-Net | 88.33 | 85.52 | 90.60 | 89.86 | 89.45 | 87.63 | 87.20 ± 3.42 | 89.85 ± 2.11 |
| | CNN + TF | 88.70 | 83.12 | 87.18 | 92.75 | 87.93 | 87.67 | 87.30 ± 3.63 | 89.67 ± 1.55 |
| | CNN + TF + DigitCapsule-Net | 86.89 | 86.21 | 90.60 | 90.58 | 88.70 | 88.34 | 87.39 ± 3.18 | 90.09 ± 2.13 |
| | SVC | 87.20 | 85.03 | 93.16 | 90.58 | 90.08 | 87.72 | 89.54 ± 2.49 | 90.66 ± 2.64 |
| | GB | 92.44 | 88.65 | 94.02 | 90.58 | 93.22 | 89.61 | 90.42 ± 2.36 | 92.18 ± 1.83 |
| CN *vs* LMCI | CNN | 92.44 | 88.00 | 94.02 | 97.35 | 93.22 | 92.44 | 89.54 ± 4.02 | 91.39 ± 3.02 |
| | CNN + DigitCapsule-Net | 92.50 | 89.26 | 94.87 | 95.58 | 93.67 | 92.31 | 90.51 ± 3.26 | 90.77 ± 3.09 |
| | CNN + TF | 91.19 | 91.19 | 91.85 | 91.85 | 91.45 | 91.45 | 89.54 ± 3.92 | 89.54 ± 3.92 |
| | CNN + TF + DigitCapsule-Net | 92.24 | 88.80 | 91.45 | 98.23 | 91.85 | 93.28 | 89.90 ± 3.50 | 91.97 ± 2.88 |
| | SVC | 94.07 | 92.50 | 94.87 | 98.23 | 94.47 | 95.28 | 91.35 ± 1.85 | 92.59 ± 2.06 |
| | GB | 95.50 | 93.97 | 90.60 | 96.46 | 92.98 | 95.20 | 91.84 ± 3.19 | 92.90 ± 1.59 |
| LMCI *vs* EMCI | CNN | 80.00 | 81.88 | 58.43 | 84.14 | 67.53 | 82.99 | 80.05 ± 3.00 | 82.92 ± 4.50 |
| | CNN + DigitCapsule-Net | 76.83 | 86.71 | 70.79 | 85.52 | 73.68 | 86.11 | 79.93 ± 3.68 | 84.37 ± 1.58 |
| | CNN + TF | 74.19 | 85.42 | 51.69 | 84.83 | 60.93 | 85.12 | 77.63 ± 4.09 | 83.74 ± 1.75 |
| | CNN + TF + DigitCapsule-Net | 77.38 | 83.92 | 73.03 | 82.76 | 75.14 | 83.33 | 78.83 ± 3.51 | 84.09 ± 2.60 |
| | SVC | 80.29 | 87.07 | 82.02 | 88.28 | 82.02 | 87.67 | 85.53 ± 2.69 | 87.60 ± 2.61 |
| | GB | 88.73 | 91.49 | 70.79 | 88.97 | 78.75 | 90.21 | 83.23 ± 6.44 | 87.77 ± 3.22 |
| AD *vs* LMCI | CNN | 40.00 | 85.85 | 40.00 | 100.0 | 40.00 | 92.39 | 83.65 ± 4.82 | 90.00 ± 3.29 |
| | CNN + DigitCapsule-Net | 52.94 | 83.33 | 36.00 | 08.90 | 42.86 | 90.45 | 91.00 ± 2.70 | 85.00 ± 4.32 |
| | CNN + TF | 59.26 | 83.33 | 64.00 | 83.33 | 61.54 | 89.00 | 83.21 ± 3.52 | 91.00 ± 2.70 |
| | CNN + TF + DigitCapsule-Net | 52.94 | 85.05 | 100.0 | 100.0 | 42.86 | 89.66 | 91.92 ± 3.52 | 91.01 ± 3.29 |
| | SVC | 69.57 | 89.22 | 64.00 | 100.0 | 66.67 | 94.30 | 84.32 ± 3.76 | 93.53 ± 2.54 |
| | GB | 58.82 | 84.62 | 40.00 | 96.70 | 47.62 | 90.26 | 82.34 ± 7.13 | 91.00 ± 3.41 |
| AD *vs* EMCI | CNN | 86.36 | 95.71 | 79.17 | 100.0 | 82.61 | 97.81 | 95.72 ± 2.35 | 97.67 ± 1.07 |
| | CNN + DigitCapsule-Net | 86.36 | 93.06 | 77.15 | 100.0 | 88.94 | 96.40 | 95.72 ± 1.35 | 98.38 ± 0.97 |
| | CNN + TF | 90.00 | 96.40 | 75.00 | 100.0 | 96.40 | 98.17 | 98.38 ± 0.97 | 98.29 ± 1.02 |
| | CNN + TF + DigitCapsule-Net | 90.91 | 95.04 | 83.33 | 100.0 | 86.96 | 97.45 | 95.41 ± 2.05 | 98.11 ± 1.10 |
| | SVC | 87.50 | 95.71 | 87.50 | 100.0 | 87.50 | 97.81 | 96.64 ± 1.49 | 98.38 ± 0.96 |
| | GB | 88.89 | 94.85 | 66.67 | 96.27 | 76.19 | 95.56 | 92.36 ± 1.49 | 95.50 ± 2.12 |
| AD *vs* CN | CNN | 92.00 | 100.0 | 95.83 | 99.19 | 93.88 | 99.59 | 98.58 ± 1.06 | 99.15 ± 1.15 |
| | CNN + DigitCapsule-Net | 88.46 | 100.0 | 95.83 | 100.0 | 92.00 | 100.0 | 98.05 ± 1.68 | 98.82 ± 1.21 |
| | CNN + TF | 88.46 | 100.0 | 95.83 | 100.0 | 92.00 | 100.0 | 98.23 ± 1.57 | 99.14 ± 0.93 |
| | CNN + TF + DigitCapsule-Net | 88.46 | 100.0 | 95.83 | 100.0 | 92.00 | 100.0 | 98.05 ± 1.23 | 99.15 ± 1.04 |

*(Continued)*

Bravo-Ortíz et al. (2025), *PeerJ Comput. Sci.*, DOI 10.7717/peerj-cs.3208

| Classes | Models | Precision [%] | | Recall [%] | | F1-score [%] | | Cross-validation [%] | |
|---|---|---|---|---|---|---|---|---|---|
| | | Imbalanced | Balanced | Imbalanced | Balanced | Imbalanced | Balanced | Imbalanced | Balanced |
| | SVC | 100.0 | 100.0 | 95.83 | 100.0 | 97.87 | 100.0 | 98.75 ± 1.14 | 99.46 ± 0.72 |
| | GB | 87.70 | 99.18 | 79.61 | 98.37 | 83.46 | 98.78 | 96.81 ± 1.90 | 99.13 ± 1.05 |

**Table 5 Results obtained from all models evaluated in three classes include precision, recall, F1-score, and the accuracy obtained in 10k-fold cross-validation.** Additionally, each result is presented for imbalanced data.

| Classes | Models | Precision [%] | Recall [%] | F1-score [%] | Cross-validation [%] |
|---|---|---|---|---|---|
| CN vs EMCI vs LMCI | CNN | 79.61 | 79.61 | 79.61 | 76.15 ± 2.13 |
| | CNN + DigitCapsule-Net | 77.62 | 79.13 | 78.37 | 76.51 ± 3.37 |
| | CNN + TF | 81.15 | 75.24 | 78.09 | 76.65 ± 2.27 |
| | CNN + TF + DigitCapsule-Net | 79.90 | 77.18 | 78.52 | 76.29 ± 2.47 |
| | ETC | 87.57 | 75.24 | 80.94 | 80.85 ± 3.34 |
| | SVC | 80.29 | 81.07 | 80.68 | 81.57 ± 3.46 |
| | RF | 86.29 | 73.30 | 79.27 | 79.76 ± 2.41 |
| | GB | 87.70 | 79.61 | 83.46 | 82.37 ± 2.86 |
| AD vs EMCI vs LMCI | CNN | 70.73 | 50.88 | 59.18 | 71.39 ± 4.63 |
| | CNN + DigitCapsule-Net | 65.38 | 59.65 | 62.39 | 74.36 ± 3.79 |
| | CNN + TF | 64.95 | 55.26 | 59.72 | 71.68 ± 4.79 |
| | CNN + TF + DigitCapsule-Net | 66.00 | 57.89 | 61.68 | 73.47 ± 4.42 |
| | ETC | 73.33 | 57.89 | 64.71 | 75.74 ± 4.87 |
| | SVC | 74.31 | 71.05 | 72.65 | 79.80 ± 3.65 |
| | RF | 72.83 | 58.77 | 65.05 | 73.46 ± 3.85 |
| | GB | 67.39 | 54.39 | 60.19 | 76.73 ± 3.71 |
| AD vs CN vs LMCI | CNN | 90.51 | 87.94 | 89.21 | 81.74 ± 2.74 |
| | CNN + DigitCapsule-Net | 86.43 | 85.82 | 86.12 | 81.74 ± 2.74 |
| | CNN + TF | 87.68 | 85.82 | 86.74 | 81.20 ± 3.15 |
| | CNN + TF + DigitCapsule-Net | 86.52 | 86.52 | 86.52 | 82.17 ± 3.08 |
| | ETC | 81.13 | 76.11 | 78.54 | 84.67 ± 1.71 |
| | SVC | 83.02 | 77.88 | 80.37 | 86.30 ± 2.75 |
| | RF | 77.68 | 76.99 | 77.33 | 84.56 ± 3.18 |
| | GB | 71.43 | 70.80 | 71.11 | 84.45 ± 2.99 |
| AD vs CN vs EMCI | CNN | 86.21 | 88.65 | 87.41 | 85.09 ± 2.80 |
| | CNN + DigitCapsule-Net | 90.97 | 92.91 | 91.93 | 90.58 ± 2.51 |
| | CNN + TF | 86.07 | 87.78 | 88.89 | 86.16 ± 2.83 |
| | CNN + TF + DigitCapsule-Net | 87.07 | 90.78 | 88.89 | 86.16 ± 2.97 |
| | ETC | 92.48 | 87.23 | 89.78 | 88.57 ± 3.33 |
| | SVC | 86.90 | 89.36 | 88.11 | 87.76 ± 2.61 |
| | RF | 93.23 | 87.94 | 90.51 | 88.75 ± 1.55 |
| | GB | 94.85 | 91.49 | 93.14 | 93.59 ± 3.13 |

## Overall results

DL models were evaluated over 500 epochs with a batch size of 256. The ML models were assessed with the hyperparameters defined in 'ML Hyperparameter Tuning with Grid Search and Pipelines'. Table 4 reports the binary-classification results and compares DL models with SVC and GB. The analysis covers precision, recall, F1-score, and 10-fold cross-validated accuracy for balanced and imbalanced datasets.

Table 5 summarizes the three-class classification results obtained with DL models and the ETC, SVC, RF, and GB. As in the binary case, performance metrics are reported for both balanced and imbalanced datasets. For metric visualization, Figures 10, 11, 12, 13 visualise the main evaluation metrics for the CN–EMCI–LMCI, AD–CN–EMCI, AD–CN–LMCI, and AD–EMCI–LMCI tasks, respectively. Each figure contrasts the two best-performing models—CNN + DigitCapsule-Net and CNN + TF. Metrics plotted comprise the classification report, confusion matrix, and receiver-operating-characteristic (ROC) curve generated with early-stopping. Finally, Table 6 shows the results using data balancing techniques.

Finally, Table 7 presents the four-class results obtained with the same ML models evaluated in the three-class scenario. Figure 14 shows the ROC curve, confusion matrix, and other metrics for the two top models—DigitCapsule-Net and the Transformer encoder.

Alzheimer's disease poses a significant global health challenge, and current projections indicate it will become an even greater concern in the medium and long term due to population aging. The growing body of research on this topic reflects its increasing relevance. However, the sheer volume of published studies complicates the identification of the most effective models and methodologies. This study evaluates and contrasts state-of-the-art and traditional approaches for classifying Alzheimer's disease, emphasizing the utility of structured clinical data, an underexplored modality often overshadowed by MRI. Advanced architectures—CNN + TF and CNN + DigitCapsule-Net—require substantially more computation time. Our findings show that, for structured clinical data, well-tuned traditional ML models can outperform complex DL models and offer superior efficiency and interpretability. State-of-the-art models (the 1-D CNN + DigitCapsule-Net) achieve competitive performance but at a substantially higher computational cost than traditional ML approaches.

Rapid AI development hampers reproducibility when source code or datasets are not publicly released. Such opacity hinders result verification and often leads to disputes over which method is superior. Moreover, many state-of-the-art techniques achieve strong headline metrics yet lack sufficient interpretability. Researchers routinely publish results but often omit the underlying rationale for their model decisions. Interpretability must therefore remain a fundamental pillar of artificial-intelligence research.

To address the class imbalance, particularly the overrepresentation of the LMCI class, we report the performance metrics for each class individually. Table 8 presents the precision, recall, and F1-score per class across all tested models without applying any balancing techniques. The LMCI class (label 3) shows consistently lower recall and F1-

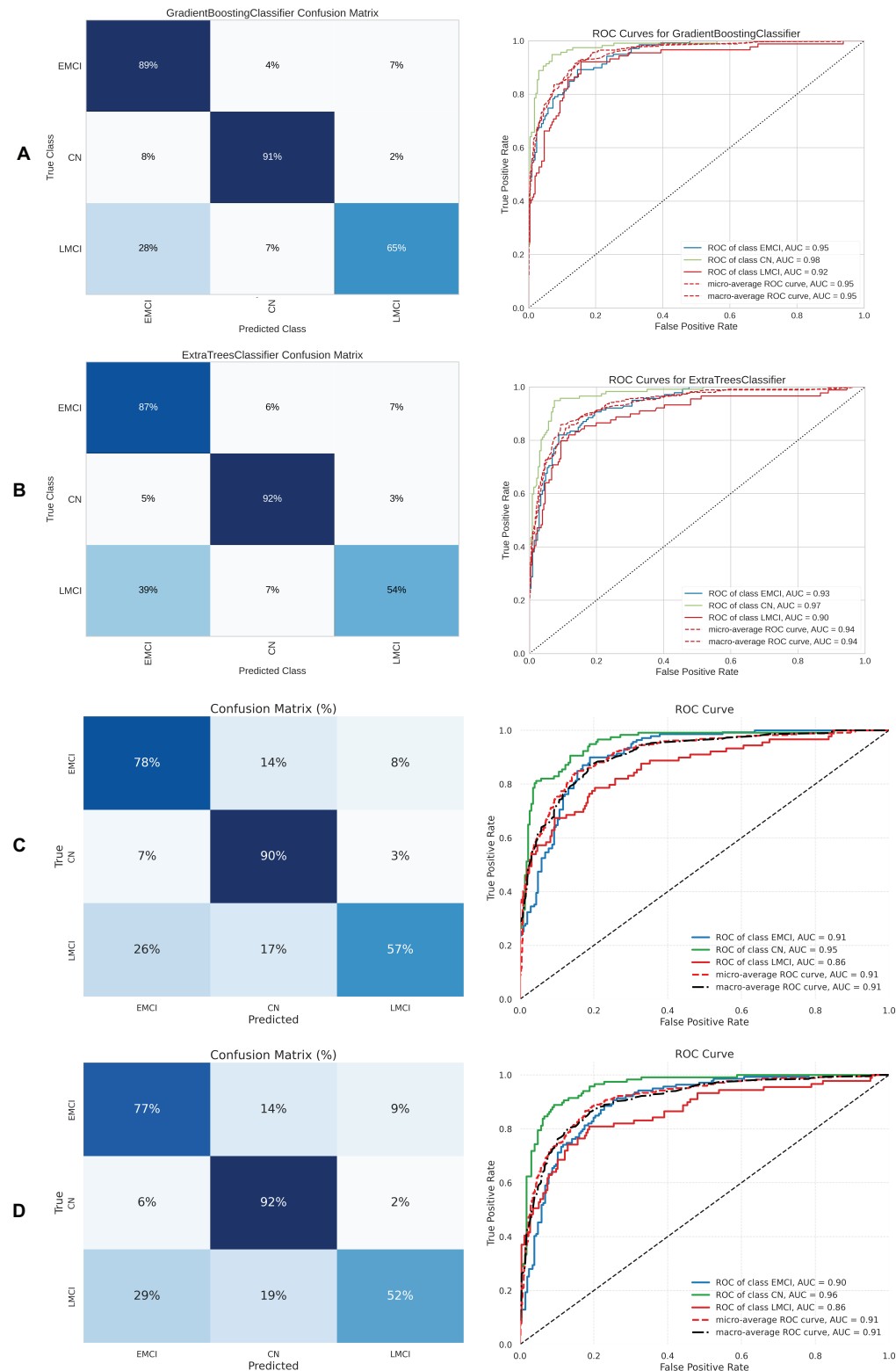

**Figure 10** **Metrics for the two best ML and DL models on an unbalanced dataset; on the right, the confusion matrix for each class (CN, EMCI, LMCI) is shown, and on the left, the ROC curve from the training of the models.** (A) Gradient boosting, (B) Extra Tree, (C) CNN + DigitCapsule-Net, (D) CNN + TF.                                       

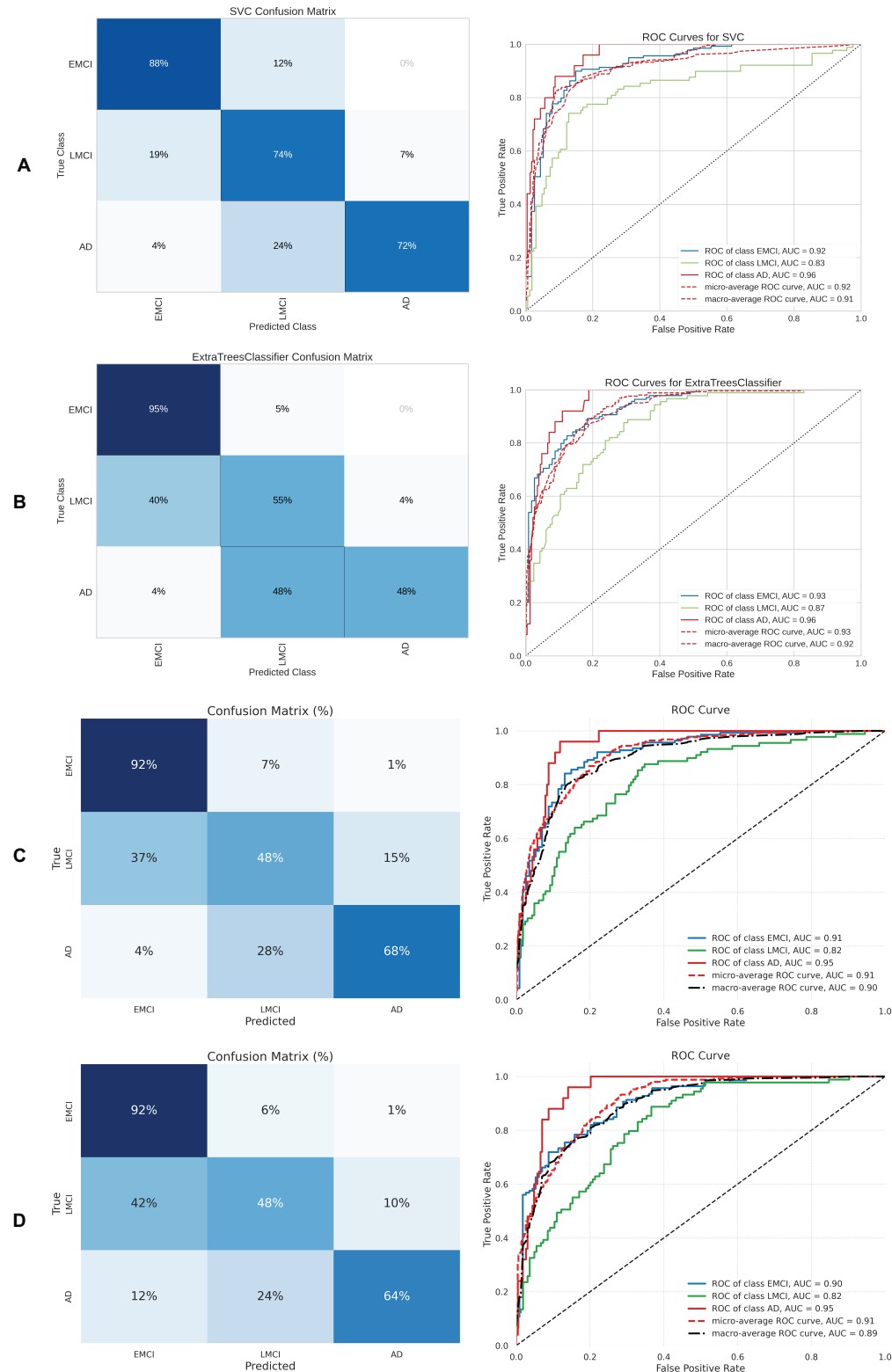

**Figure 11 Metrics for the two best ML and DL models on an unbalanced dataset; on the right, the confusion matrix for each class (AD, EMCI, LMCI) is shown, and on the left, the ROC curve from the training of the models.** (A) SVM, (B) Extra Tree, (C) CNN + DigitCapsule-Net, (D) CNN + TF.

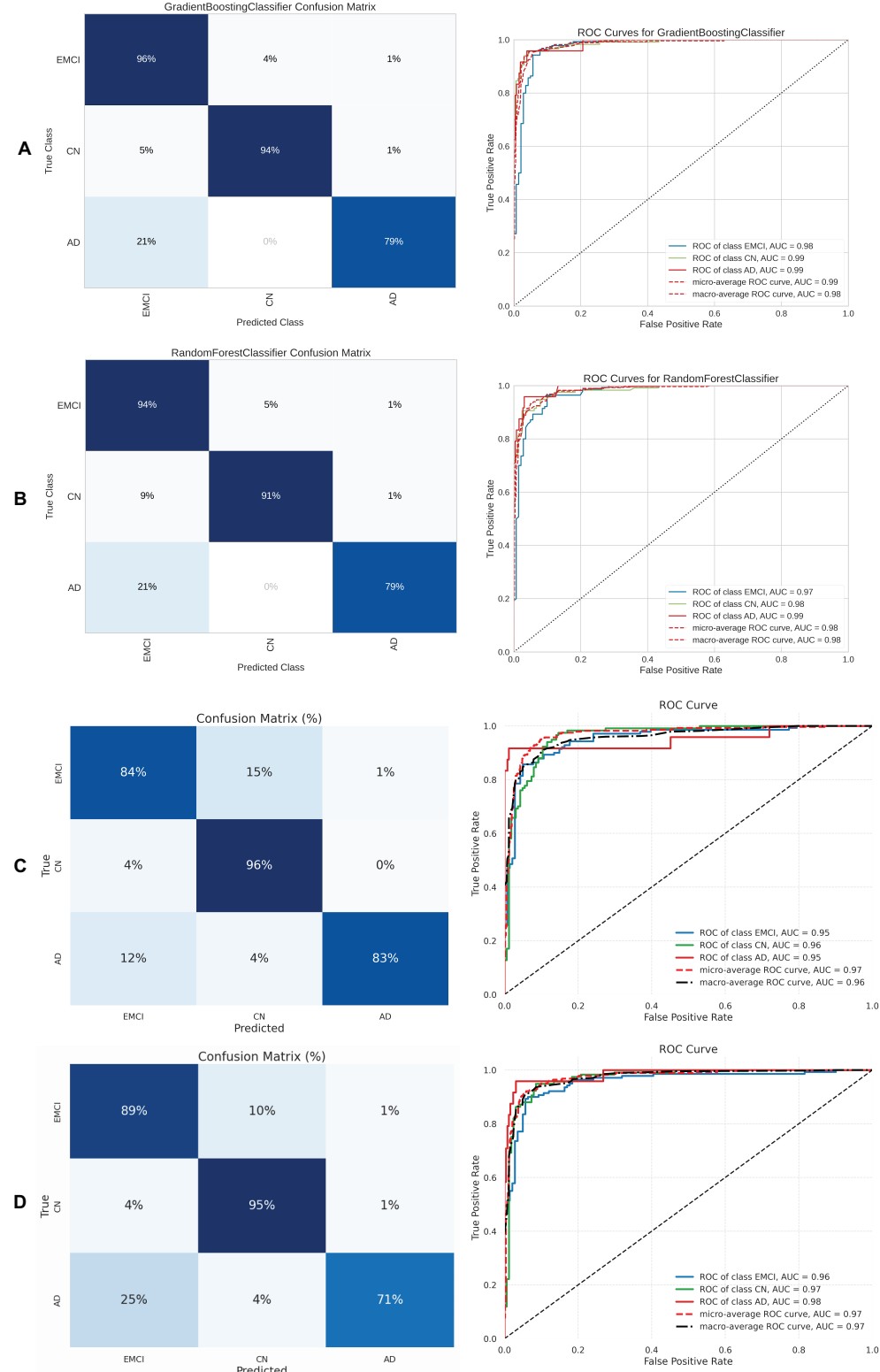

**Figure 12** **Metrics for the two best ML and DL models on an unbalanced dataset; on the right, the confusion matrix for each class (CN, EMCI, AD) is shown, and on the left, the ROC curve from the training of the models.** (A) Gradient boosting, (B) Extra Tree, (C) CNN + DigitCapsule-Net, (D) CNN + TF.

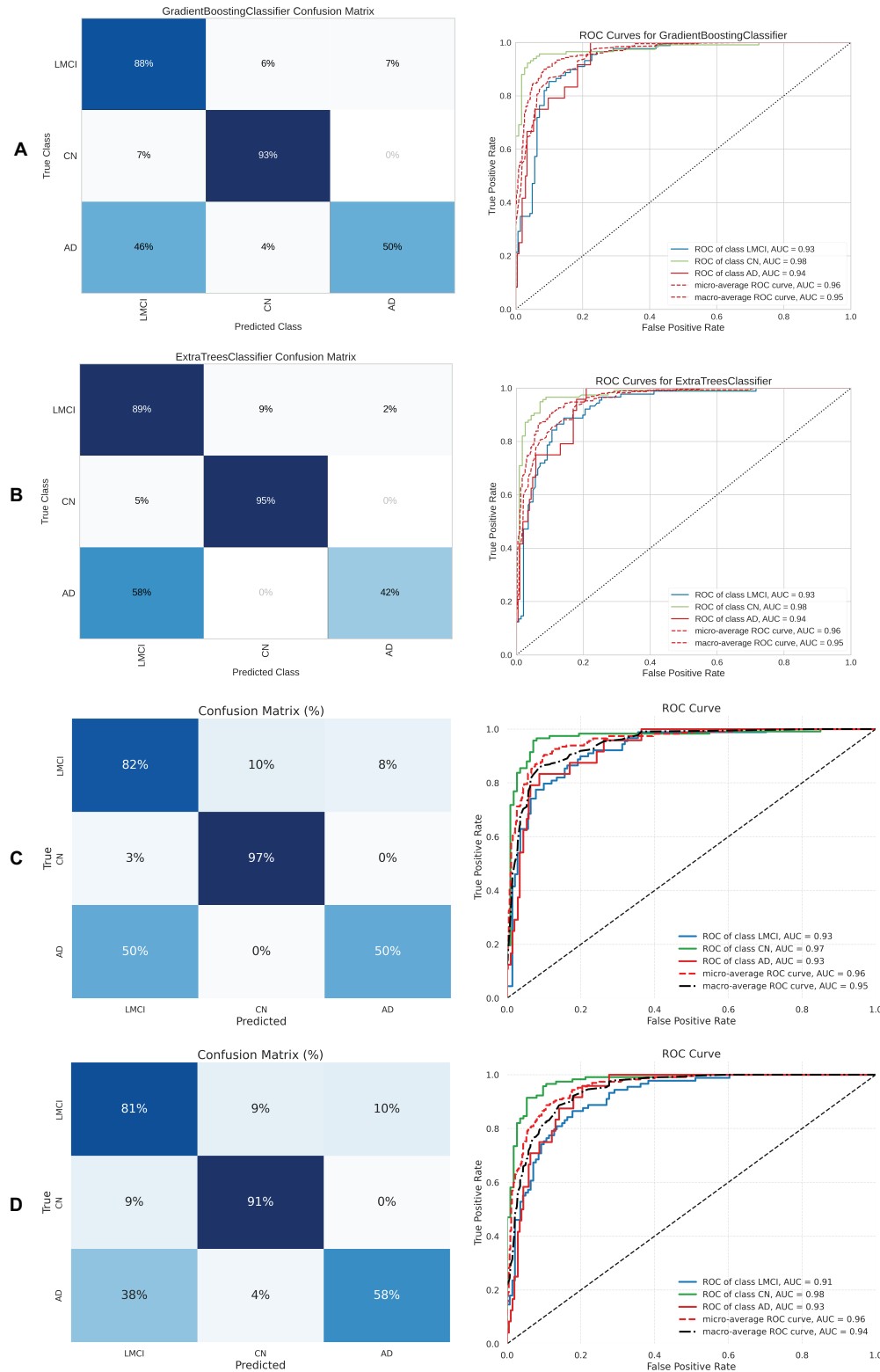

**Figure 13  Metrics for the two best ML and DL models on an unbalanced dataset; on the right, the confusion matrix for each class (CN, LMCI, AD) is shown, and on the left, the ROC curve from the training of the models.** (A) Gradient boosting, (B) Extra Tree, (C) CNN + DigitCapsule-Net, (D) CNN + TF.

**Table 6 Results obtained from all models evaluated in three classes.** Each result includes precision, recall, F1-score, and accuracy obtained in 10k-fold cross-validation for different balancing methods (SMOTE, SMOTE-Tomek, ADASYN, and NearMiss).

| Classes | Models | Precision [%] | | | | Recall [%] | | | | F1-score [%] | | | | Cross-validation [%] | | | |
|---|---|---|---|---|---|---|---|---|---|---|---|---|---|---|---|---|---|
| | | SMOTE | SMOTE-Tomek | ADASYN | NearMiss | SMOTE | SMOTE-Tomek | ADASYN | NearMiss | SMOTE | SMOTE-Tomek | ADASYN | NearMiss | SMOTE | SMOTE-Tomek | ADASYN | NearMiss |
| CN vs EMCI vs LMCI | CNN | 81.16 | 80.88 | 83.25 | 76.14 | 86.50 | 83.88 | 80.48 | 77.01 | 83.75 | 82.35 | 81.84 | 76.57 | 80.72 ± 3.36 | 75.73 ± 2.19 | 47.18 ± 6.67 | 77.62 ± 2.67 |
| | CNN + DigitCapsule-Net | 81.56 | 80.08 | 81.41 | 79.29 | 83.94 | 84.71 | 77.14 | 77.01 | 82.73 | 82.33 | 79.22 | 78.13 | 81.50 ± 2.15 | 77.06 ± 1.97 | 74.11 ± 3.64 | 77.06 ± 2.07 |
| | CNN + TF | 82.91 | 82.38 | 80.42 | 78.26 | 83.21 | 83.06 | 72.38 | 72.41 | 83.06 | 82.72 | 76.19 | 75.22 | 81.62 ± 3.03 | 75.87 ± 2.37 | 73.62 ± 4.22 | 76.50 ± 2.39 |
| | CNN + TF + DigitCapsule-Net | 79.66 | 80.39 | 80.00 | 77.78 | 85.77 | 84.71 | 80.00 | 76.44 | 82.60 | 82.49 | 80.00 | 77.10 | 80.84 ± 2.98 | 77.06 ± 2.72 | 72.85 ± 3.35 | 76.50 ± 4.00 |
| | ETC | 85.00 | 86.00 | 81.00 | 81.00 | 86.86 | 86.00 | 80.00 | 81.00 | 85.92 | 86.00 | 80.00 | 81.00 | 83.59 ± 2.82 | 85.00 ± 2.00 | 81.00 ± 3.00 | 81.00 ± 3.00 |
| | SVC | 89.05 | 85.00 | 81.00 | 79.00 | 89.05 | 85.00 | 81.00 | 80.00 | 89.05 | 85.00 | 81.00 | 79.00 | 84.91 ± 3.45 | 85.00 ± 3.00 | 82.00 ± 3.00 | 82.00 ± 2.00 |
| | RF | 84.23 | 84.00 | 82.00 | 79.00 | 85.77 | 84.00 | 81.00 | 79.00 | 84.99 | 84.00 | 81.00 | 79.00 | 82.39 ± 2.93 | 84.00 ± 3.00 | 79.00 ± 3.00 | 80.00 ± 2.00 |
| | GB | 89.64 | 85.00 | 84.00 | 78.00 | 91.61 | 85.00 | 84.00 | 77.00 | 90.61 | 85.00 | 84.00 | 78.00 | 85.14 ± 2.31 | 85.00 ± 2.00 | 82.00 ± 3.00 | 83.00 ± 2.00 |
| AD vs EMCI vs LMCI | CNN | 86.27 | 85.60 | 82.35 | 86.27 | 85.96 | 79.55 | 72.54 | 85.96 | 85.96 | 82.47 | 77.13 | 86.12 | 84.67 ± 2.72 | 83.78 ± 3.51 | 79.70 ± 3.34 | 66.80 ± 8.25 |
| | CNN + DigitCapsule-Net | 84.84 | 87.71 | 82.35 | 67.44 | 82.46 | 76.95 | 72.54 | 67.44 | 83.63 | 81.98 | 77.13 | 67.44 | 83.05 ± 3.21 | 83.59 ± 3.48 | 79.86 ± 3.14 | 66.80 ± 8.10 |
| | CNN + TF | 84.15 | 85.65 | 85.71 | 57.14 | 83.86 | 75.46 | 68.39 | 65.12 | 84.01 | 80.24 | 76.08 | 60.87 | 83.41 ± 1.97 | 83.33 ± 2.75 | 79.94 ± 2.79 | 66.83 ± 8.49 |
| | CNN + TF + DigitCapsule-Net | 85.37 | 90.39 | 83.65 | 64.71 | 85.96 | 76.95 | 68.91 | 76.74 | 85.66 | 83.13 | 75.57 | 70.21 | 83.29 ± 1.81 | 83.33 ± 2.39 | 80.09 ± 2.62 | 65.78 ± 6.89 |
| | ETC | 90.36 | 88.00 | 83.00 | 74.00 | 88.77 | 89.00 | 80.00 | 75.00 | 89.56 | 88.00 | 79.00 | 74.00 | 86.64 ± 3.13 | 87.00 ± 2.00 | 83.00 ± 2.00 | 68.00 ± 6.00 |
| | SVC | 91.81 | 91.00 | 87.00 | 70.00 | 90.53 | 91.00 | 86.00 | 69.00 | 91.17 | 91.00 | 86.00 | 68.00 | 88.92 ± 1.91 | 89.00 ± 2.00 | 85.00 ± 2.00 | 73.00 ± 7.00 |
| | RF | 87.46 | 87.00 | 80.00 | 71.00 | 85.61 | 88.00 | 77.00 | 70.00 | 86.52 | 87.00 | 76.00 | 69.00 | 84.01 ± 2.55 | 84.00 ± 2.00 | 81.00 ± 2.00 | 70.00 ± 6.00 |
| | GB | 90.11 | 89.00 | 84.00 | 70.00 | 89.47 | 89.00 | 83.00 | 70.00 | 89.79 | 89.00 | 83.00 | 68.00 | 88.62 ± 2.46 | 86.00 ± 3.00 | 85.00 ± 2.00 | 72.00 ± 7.00 |
| AD vs CN vs LMCI | CNN | 89.20 | 89.29 | 88.52 | 67.86 | 95.30 | 95.69 | 96.86 | 88.37 | 92.15 | 92.38 | 92.51 | 76.77 | 89.14 ± 1.88 | 88.85 ± 3.28 | 86.72 ± 2.47 | 73.70 ± 10.31 |
| | CNN + DigitCapsule-Net | 87.31 | 87.17 | 86.80 | 73.08 | 97.01 | 94.26 | 97.31 | 88.37 | 91.90 | 90.57 | 91.75 | 80.00 | 89.92 ± 1.08 | 90.97 ± 3.55 | 87.05 ± 1.94 | 77.07 ± 8.63 |
| | CNN + TF | 88.89 | 87.95 | 87.10 | 77.08 | 95.73 | 94.26 | 96.86 | 86.05 | 92.18 | 90.99 | 91.72 | 81.32 | 89.78 ± 1.60 | 90.42 ± 3.08 | 86.88 ± 1.80 | 76.74 ± 9.71 |
| | CNN + TF + DigitCapsule-Net | 87.98 | 89.04 | 87.70 | 76.00 | 97.01 | 93.30 | 99.10 | 88.37 | 92.28 | 91.12 | 93.05 | 81.72 | 89.42 ± 1.71 | 89.87 ± 2.95 | 87.46 ± 1.20 | 75.70 ± 6.16 |
| | ETC | 89.33 | 89.00 | 92.00 | 85.00 | 89.33 | 89.00 | 92.00 | 86.00 | 89.33 | 89.00 | 92.00 | 85.00 | 90.34 ± 3.09 | 91.00 ± 3.00 | 89.00 ± 2.00 | 82.00 ± 7.00 |
| | SVC | 92.20 | 92.00 | 91.00 | 79.00 | 89.33 | 92.00 | 90.00 | 74.00 | 90.74 | 92.00 | 90.00 | 76.00 | 92.56 ± 3.07 | 93.00 ± 3.00 | 91.00 ± 2.00 | 82.00 ± 7.00 |
| | RF | 85.33 | 87.00 | 91.00 | 86.00 | 85.33 | 88.00 | 90.00 | 86.00 | 85.33 | 87.00 | 90.00 | 85.00 | 89.91 ± 3.53 | 90.00 ± 4.00 | 89.00 ± 3.00 | 83.00 ± 8.00 |
| | GB | 89.29 | 87.00 | 91.00 | 83.00 | 88.89 | 88.00 | 91.00 | 83.00 | 89.09 | 87.00 | 91.00 | 83.00 | 90.06 ± 3.44 | 91.00 ± 2.00 | 88.00 ± 3.00 | 83.00 ± 8.00 |
| AD vs CN vs EMCI | CNN | 94.10 | 91.67 | 89.82 | 79.59 | 96.96 | 95.65 | 92.27 | 90.70 | 95.51 | 93.62 | 91.03 | 84.78 | 91.38 ± 1.88 | 91.85 ± 1.61 | 89.53 ± 2.56 | 78.45 ± 6.78 |
| | CNN + DigitCapsule-Net | 93.85 | 94.85 | 90.87 | 73.58 | 97.97 | 96.27 | 95.43 | 90.70 | 95.87 | 95.56 | 93.10 | 81.25 | 92.66 ± 2.43 | 92.18 ± 1.79 | 90.51 ± 3.05 | 81.86 ± 6.76 |
| | CNN + TF | 92.01 | 94.09 | 89.57 | 78.43 | 97.30 | 94.47 | 93.64 | 93.02 | 94.58 | 94.28 | 91.56 | 85.11 | 91.68 ± 1.27 | 90.64 ± 1.59 | 89.17 ± 2.88 | 78.78 ± 7.10 |
| | CNN + TF + DigitCapsule-Net | 94.68 | 93.44 | 89.27 | 76.92 | 96.28 | 95.65 | 94.55 | 93.02 | 95.48 | 94.53 | 91.83 | 84.21 | 90.90 ± 2.62 | 92.11 ± 1.69 | 90.52 ± 2.47 | 77.39 ± 7.99 |
| | ETC | 95.27 | 92.00 | 92.00 | 86.00 | 95.27 | 92.00 | 92.00 | 87.00 | 95.27 | 92.00 | 92.00 | 87.00 | 92.75 ± 1.86 | 93.00 ± 1.00 | 92.00 ± 3.00 | 91.00 ± 5.00 |
| | SVC | 92.83 | 93.00 | 90.00 | 86.00 | 96.28 | 93.00 | 90.00 | 88.00 | 94.53 | 93.00 | 90.00 | 87.00 | 91.43 ± 2.20 | 92.00 ± 3.00 | 91.00 ± 2.00 | 84.00 ± 6.00 |
| | RF | 95.25 | 93.00 | 92.00 | 83.00 | 94.93 | 93.00 | 92.00 | 82.00 | 95.09 | 93.00 | 92.00 | 82.00 | 92.51 ± 2.06 | 93.00 ± 1.00 | 91.00 ± 4.00 | 89.00 ± 7.00 |
| | GB | 95.67 | 94.00 | 91.00 | 82.00 | 96.96 | 94.00 | 92.00 | 83.00 | 96.31 | 94.00 | 91.50 | 82.00 | 92.87 ± 2.13 | 93.00 ± 2.00 | 92.00 ± 3.00 | 88.00 ± 7.00 |

**Table 7 Results obtained from all models evaluated in four classes include precision, recall, F1-score, and the accuracy obtained in 10k-fold cross-validation.** Additionally, each result is presented for both imbalanced and balanced data using SMOTE method.

| Classes | Models | Precision [%] | | Recall [%] | | F1-score [%] | | Cross-validation [%] | |
|---|---|---|---|---|---|---|---|---|---|
| | | Imbalanced | Balanced | Imbalanced | Balanced | Imbalanced | Balanced | Imbalanced | Balanced |
| AD *vs* CN *vs* EMCI *vs* LMCI | CNN | 73.89 | 84.94 | 72.61 | 85.95 | 73.25 | 85.44 | 72.70 ± 2.65 | 82.67 ± 1.48 |
| | CNN + DigitCapsule-Net | 71.19 | 82.60 | 73.04 | 84.76 | 72.10 | 83.67 | 73.91 ± 3.65 | 80.83 ± 0.95 |
| | CNN + TF | 71.18 | 80.32 | 70.87 | 82.62 | 71.02 | 81.46 | 73.99 ± 3.66 | 81.77 ± 1.73 |
| | CNN + TF + DigitCapsule-Net | 71.98 | 81.53 | 72.61 | 86.19 | 72.29 | 83.80 | 73.58 ± 2.97 | 82.26 ± 2.18 |
| | ETC | 81.25 | 86.97 | 73.48 | 87.38 | 77.17 | 87.17 | 75.87 ± 3.07 | 84.68 ± 1.83 |
| | SVC | 75.11 | 86.01 | 76.09 | 89.29 | 75.59 | 87.62 | 77.36 ± 3.17 | 86.43 ± 2.57 |
| | RF | 81.07 | 86.73 | 72.61 | 84.05 | 76.61 | 85.37 | 75.94 ± 2.70 | 83.79 ± 1.99 |
| | GB | 79.17 | 88.89 | 74.35 | 87.62 | 76.68 | 88.25 | 77.97 ± 3.89 | 85.45 ± 1.53 |

score values compared to the CN and EMCI classes, indicating that the models are more challenged when identifying LMCI cases. Despite this, all models demonstrate reasonably balanced performance, and the macro-averaged metrics (also shown) help mitigate the influence of class imbalance. Furthermore, this per-class analysis provides a clearer understanding of how each architecture handles underrepresented classes, as recommended by the reviewer.

## Statistical tests

Statistical significance was assessed with non-parametric and variance-based tests (Table 9), in line with best practices for ML model comparison.

A Friedman test compared multiple classifiers across cross-validation folds for experiments that employed SMOTE-Tomek resampling. For the *AD-CN-EMCI* task, the test produced a $\chi^2 = 2.89$ ($p = 0.23$), indicating no statistically significant differences among the models. Similarly, for the *CN-EMCI-LMCI* task, the Friedman test produced a result of $\chi^2 = 1.11$, $p = 0.57$, suggesting no significant differences. Levene's test showed $p$-values > 0.4 across SMOTE-Tomek tasks, indicating no statistically significant difference. A pairwise Wilcoxon signed-rank test between ETC and GB for the second task yielded a $p$-value of 0.98, reinforcing the absence of significant differences.

In ADASYN experiments, pairwise Wilcoxon tests compared each pair of classifiers. For *AD-EMCI-LMCI*, *CN-EMCI-LMCI*, and *AD-CN-EMCI* tasks, all $p$-values exceeded 0.79, again indicating no significant differences. Levene's tests also confirmed homogeneity of variance (all $p > 0.5$).

The NearMiss resampling experiment on the *AD-CN-LMCI* task included a Wilcoxon signed-rank test comparing RF and GB, yielding a $p$-value of 1.0000. Levene's test showed no significant variance differences ($p = 0.51$).

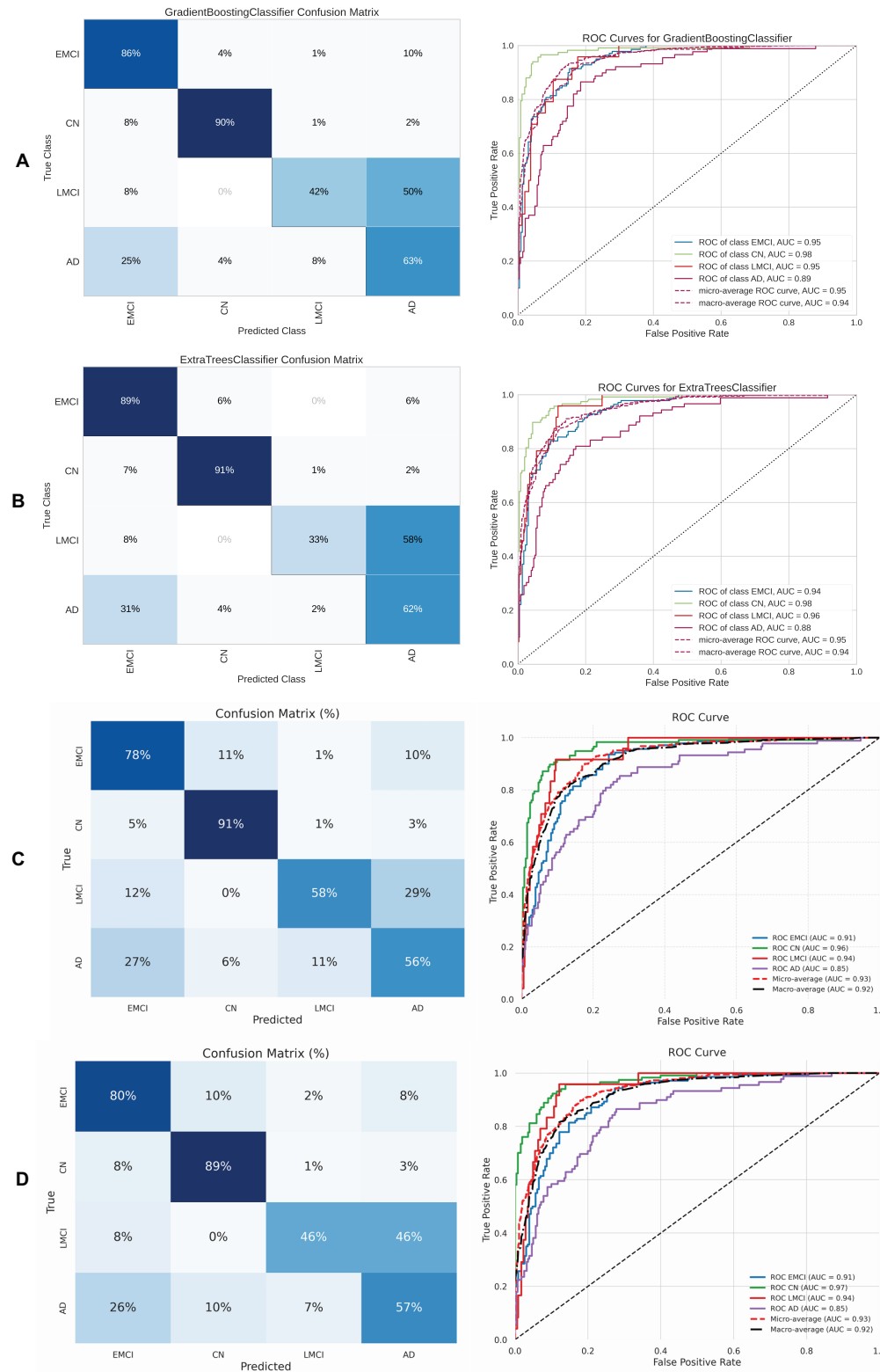

**Figure 14** **Metrics for the two best ML and DL models on an unbalanced dataset; on the right, the confusion matrix for each class (CN, MCI, LMCI, AD) is shown, and on the left, the ROC curve from the training of the models.** (A) Gradient boosting, (B) Extra Tree, (C) CNN + DigitCapsule-Net, (D) CNN + TF.

**Table 8 Per-class performance metrics (Precision, Recall, F1-score) for all evaluated models without applying class balancing.** Labels: 0 = EMCI, 1 = CN, 2 = AD, 3 = LMCI.

| Model | Class Label | Precision [%] | Recall [%] | F1-score [%] | Accuracy [%] |
|---|---|---|---|---|---|
| CNN + DigitCapsule-Net | 0 | 77.0 | 78.0 | 77.0 | 75.0 |
| | 1 | 83.0 | 91.0 | 87.0 | |
| | 2 | 54.0 | 58.0 | 56.0 | |
| | 3 | 67.0 | 56.0 | 61.0 | |
| CNN + DigitCapsule-Net + TF | 0 | 76.8 | 75.7 | 76.3 | 73.8 |
| | 1 | 80.9 | 90.6 | 85.5 | |
| | 2 | 52.2 | 50.0 | 51.1 | |
| | 3 | 62.8 | 55.1 | 58.7 | |
| CNN | 0 | 74.2 | 70.0 | 72.1 | 71.1 |
| | 1 | 78.5 | 90.6 | 84.1 | |
| | 2 | 52.4 | 45.8 | 48.9 | |
| | 3 | 58.5 | 53.9 | 56.1 | |
| CNN + TF | 0 | 77.0 | 80.0 | 78.0 | 75.0 |
| | 1 | 82.0 | 89.0 | 85.0 | |
| | 2 | 52.0 | 46.0 | 49.0 | |
| | 3 | 67.0 | 57.0 | 62.0 | |
| ETC | 0 | 77.0 | 89.0 | 82.0 | 79.2 |
| | 1 | 90.0 | 91.0 | 90.0 | |
| | 2 | 73.0 | 33.0 | 46.0 | |
| | 3 | 70.0 | 62.0 | 65.0 | |
| SVC | 0 | 79.0 | 77.0 | 78.0 | 76.5 |
| | 1 | 81.0 | 89.0 | 85.0 | |
| | 2 | 69.0 | 46.0 | 55.0 | |
| | 3 | 67.0 | 67.0 | 67.0 | |
| RF | 0 | 76.0 | 89.0 | 82.0 | 78.9 |
| | 1 | 92.0 | 90.0 | 91.0 | |
| | 2 | 64.0 | 38.0 | 47.0 | |
| | 3 | 68.0 | 60.0 | 63.0 | |
| GB | 0 | 78.0 | 86.0 | 82.0 | 78.6 |
| | 1 | 92.0 | 90.0 | 91.0 | |
| | 2 | 53.0 | 42.0 | 47.0 | |
| | 3 | 67.0 | 63.0 | 65.0 | |

## Interpretation

Correlations among feature extraction layers were analyzed to enhance model interpretability. For each layer, filter outputs were summed and averaged. This procedure was applied to data from five patients, grouped into two classes, to evaluate cross-correlations and measure the similarity between the one-dimensional convolutional signal sequences. The study analyzed layers containing eight, 64, and 256 filters and applied the same averaging method to calculate the Euclidean distance between the convolutional output signals, as shown in Table 10.

**Table 9 Statistical comparison of model performances across different class-balancing techniques and classification tasks.**

| Resampling | Task | Test | Models compared | Statistic | *p*-value |
|---|---|---|---|---|---|
| SMOTE-Tomek | AD *vs* CN *vs* EMCI | Friedman | ET, RF, GB | 2.8889 | 0.2359 |
| | | Levene (variance) | ET, RF, GB | 0.9076 | 0.4155 |
| | CN *vs* EMCI *vs* LMCI | Friedman | ET, SVM, GB | 1.1176 | 0.5719 |
| | | Levene (variance) | ET, SVM, GB | 0.2149 | 0.8080 |
| | | Wilcoxon | ET *vs* GB | 17.5000 | 0.9844 |
| ADASYN | AD *vs* EMCI *vs* LMCI | Wilcoxon | SVM *vs* GB | 24.5000 | 0.7949 |
| | | Levene (variance) | SVM, GB | 0.3785 | 0.5461 |
| | CN *vs* EMCI *vs* LMCI | Wilcoxon | SVM *vs* GB | 22.0000 | 1.0000 |
| | | Levene (variance) | SVM, GB | 0.1761 | 0.6798 |
| | AD *vs* CN *vs* EMCI | Wilcoxon | ET *vs* GB | 17.5000 | 0.9766 |
| | | Levene (variance) | ET, GB | 0.2327 | 0.6354 |
| NearMiss | AD *vs* CN *vs* LMCI | Wilcoxon | RF *vs* GB | 10.5000 | 1.0000 |
| | | Levene (variance) | RF, GB | 0.4358 | 0.5175 |

**Table 10 Correlation of the average obtained in each filter of the feature extraction layers in five different patients, in the convolutions 8, 64, and 256, is achieved by calculating the cross-correlation.** The L1 column, referring to the Euclidean distance, is computed using the same procedure.

| Classes | ID | Correlation | | | L1 | | |
|---|---|---|---|---|---|---|---|
| | | 8 | 64 | 256 | 8 | 64 | 256 |
| CN *vs* EMCI | A | 0.4030 | 0.3020 | 0.4907 | 2.8712 | 1.5299 | 0.3458 |
| | B | 0.5443 | 0.5062 | 0.5268 | 2.6252 | 1.4302 | 0.4262 |
| | C | −0.3037 | −0.0385 | 0.1390 | 5.7895 | 2.4832 | 0.6328 |
| | D | −0.0581 | −0.1191 | 0.1707 | 4.0957 | 2.0928 | 0.6009 |
| | E | 0.2618 | 0.0621 | 0.1969 | 2.9074 | 1.6487 | 0.4686 |
| CN *vs* LMCI | A | −0.4301 | 0.0399 | 0.1868 | 5.5092 | 2.2016 | 0.5690 |
| | B | −0.4655 | −0.2083 | −0.1937 | 4.9820 | 2.2796 | 0.7109 |
| | C | −0.0819 | −0.2042 | −0.0378 | 5.6587 | 2.7651 | 0.7637 |
| | D | 0.6961 | 0.6251 | 0.3842 | 2.4520 | 1.3135 | 0.4291 |
| | E | −0.3286 | −0.1685 | −0.1337 | 4.5221 | 2.2976 | 0.6447 |
| CN *vs* AD | A | −0.4928 | −0.2573 | 0.1400 | 3.8448 | 1.8343 | 0.4198 |
| | B | −0.6169 | −0.1534 | 0.0003 | 6.5764 | 2.5505 | 0.7084 |
| | C | 0.5925 | 0.5026 | 0.4523 | 4.3721 | 1.9843 | 0.6180 |
| | D | −0.5753 | −0.2602 | −0.0024 | 6.7331 | 2.7062 | 0.6994 |
| | E | −0.3856 | 0.0864 | 0.3709 | 5.4038 | 2.0716 | 0.5531 |
| EMCI *vs* LMCI | A | −0.6768 | −0.1865 | 0.0487 | 6.7987 | 2.5821 | 0.6434 |
| | B | −0.3794 | −0.1835 | −0.0336 | 4.8073 | 2.3412 | 0.6770 |
| | C | −0.0171 | 0.0921 | 0.0613 | 4.4338 | 2.0839 | 0.6297 |
| | D | 0.1130 | −0.0498 | 0.1910 | 3.4711 | 2.0071 | 0.5520 |
| | E | −0.1798 | 0.0159 | −0.0294 | 4.9595 | 2.3868 | 0.7019 |

| Table 10 (continued) | | | | | | | |
|---|---|---|---|---|---|---|---|
| Classes | ID | Correlation | | | L1 | | |
| | | 8 | 64 | 256 | 8 | 64 | 256 |
| EMCI *vs* AD | A | −0.5119 | −0.0811 | 0.1106 | 4.7313 | 1.9380 | 2.5821 |
| | B | −0.4125 | −0.1681 | 0.0950 | 6.1726 | 2.7010 | 2.3412 |
| | C | −0.1049 | −0.0575 | 0.0162 | 6.4806 | 2.7911 | 2.0839 |
| | D | −0.2733 | −0.0876 | 0.1288 | 5.4382 | 2.4167 | 2.0071 |
| | E | −0.6898 | −0.1976 | 0.0605 | 6.5576 | 2.5529 | 2.3868 |
| LMCI *vs* AD | A | 0.6467 | 0.5661 | 0.5399 | 3.3416 | 1.6508 | 0.4404 |
| | B | 0.7448 | 0.6593 | 0.5924 | 3.1548 | 1.5380 | 0.4413 |
| | C | 0.3217 | 0.2867 | 0.2637 | 5.1736 | 2.2227 | 0.6261 |
| | D | −0.6751 | −0.2321 | 0.1125 | 6.6840 | 2.7333 | 0.6366 |
| | E | −0.0970 | −0.0995 | 0.1337 | 5.9284 | 2.8743 | 0.7851 |

Feature importance was derived using a GB to examine the model's behavior further, which consistently yielded superior performance. This analysis was conducted for the three-class classification tasks using imbalanced and balanced datasets, as illustrated in Fig. 15.

To understand the underlying mechanisms of the proposed model, it is essential to identify patterns that cluster similar classes and distinguish dissimilar ones. The Euclidean distance and cross-correlation analyses reveal consistent patterns among data within the same group, even when sourced from different patients. These observations support the conclusion that the feature extraction component of the model performs effectively. As emphasized earlier, elucidating the model's decision-making process is crucial, as reporting only performance metrics is insufficient.

Figure 15 hows that the most influential features vary with dementia subtype and dataset balance. This variability implies that balancing techniques can bias feature importance interpretations. Although balancing methods can stabilise models, they may obscure the underlying data behaviour.

Nevertheless, feature 11 (CDRSB, Clinical Dementia Rating–Sum of Boxes Score) and feature 19 (mPACCtrailsB, Modified Preclinical Alzheimer Cognitive Composite with Trails Test) consistently emerged as the most critical variables across conditions. This finding aligns with existing clinical research, highlighting the significance of these features in the classification of dementia (*Donohue et al., 2014*, *O'Bryant et al., 2008*).

## GradCam

Grad-CAM visualises which parts of the one-dimensional input most influence decisions made by the CNN + DigitCapsule-Net. Figure 16 presents Grad-CAM visualizations for various patient groups—CN, EMCI, LMCI, and AD—across multiple convolutional layers (8, 16, 32, 64, 128, 256). These layers are integral to the feature extraction process, during which the CNN+DigitCapsule-Net learns and identifies salient patterns within the 1D

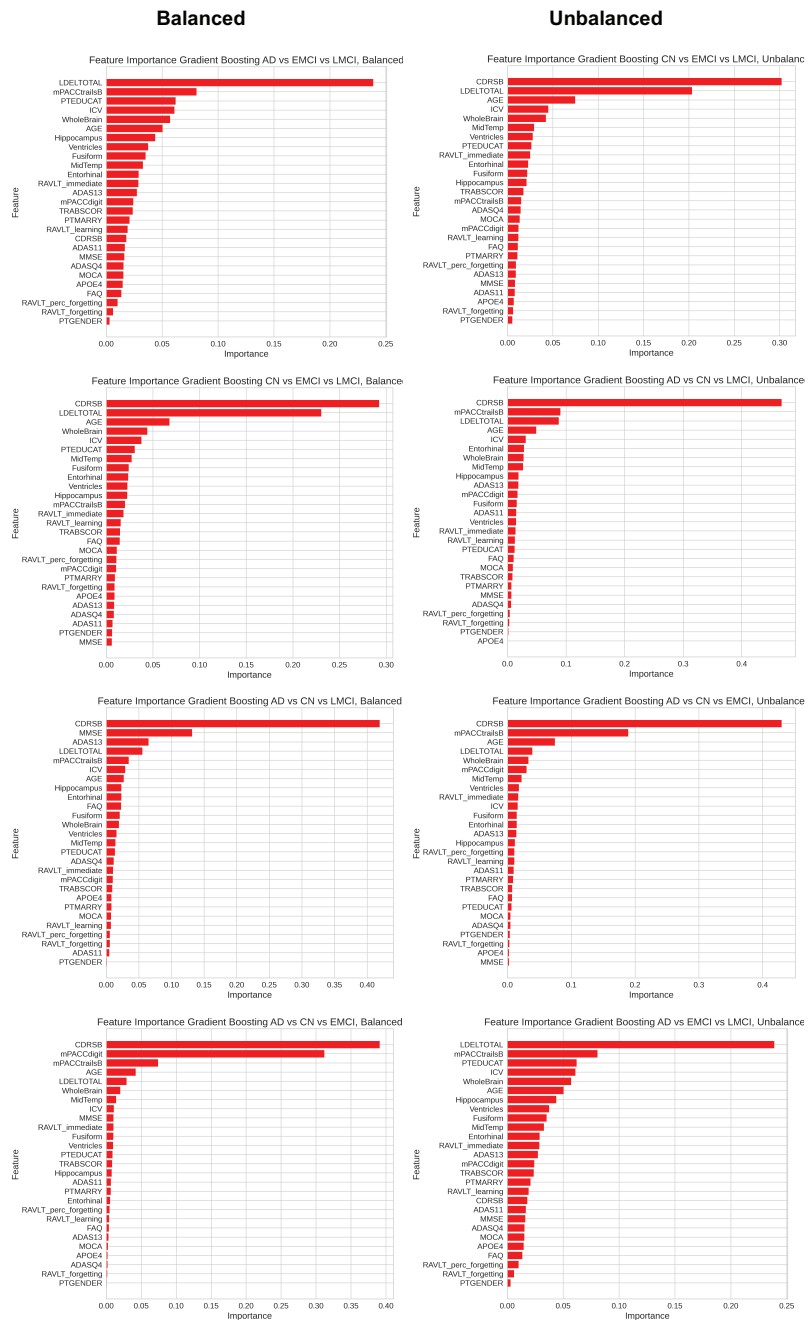

**Figure 15 Feature importance plot obtained from the GB model for each class using balanced and imbalanced data.** The y-axis represents each of the 27 features, whose identifiers are specified in Table 1.

input signals. Red regions in the visualizations highlight areas that are highly relevant to the classification outcome, whereas blue regions denote areas of lower importance. The earlier layers (8, 16) typically capture low-level features, the intermediate layers (32, 64) detect more complex structures, and the deeper layers (128, 256) represent high-level, abstract features that are more discriminative for classification. The Grad-CAM

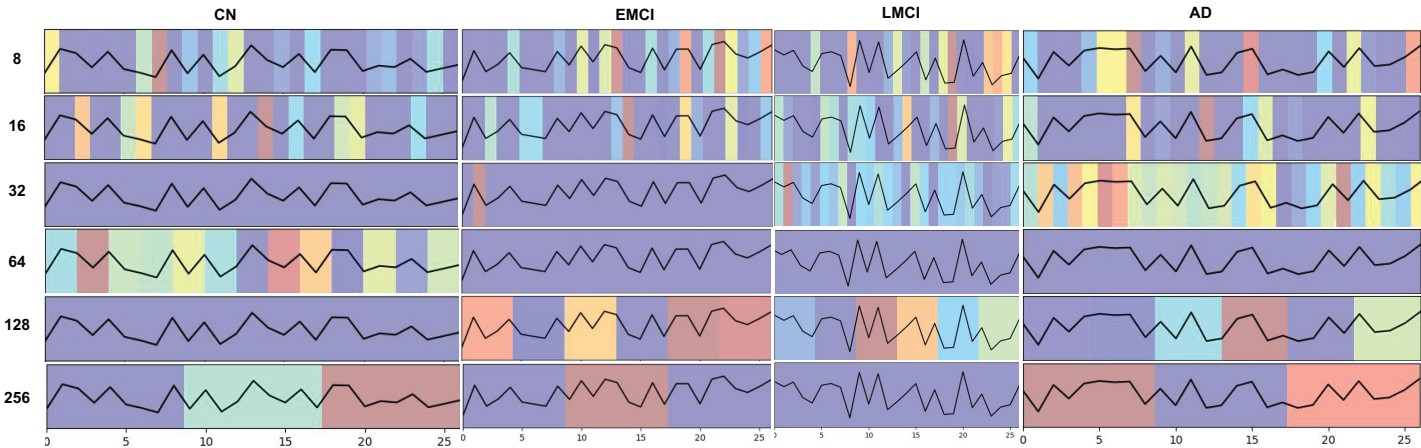

**Figure 16** GradCam visualization for each patient type in the database used in this study (CN, EMCI, LMCI, AD) at each of the feature extraction convolution layers (8, 16, 32, 64, 128, 256) of CNN+DigitCapsule-Net.

visualizations in Fig. 16 elucidate the internal representations learned at various stages of the network and provide critical insight into how the CNN+DigitCapsule-Net distinguishes between different stages of cognitive impairment and normal controls. This interpretability is essential for optimizing the model's performance and ensuring its reliability in clinical applications.

## DISCUSSION

AD is an increasingly pressing global health challenge driven by worldwide population aging. Although MRI has long been the predominant modality in AI-based AD research, its high cost and limited availability restrict widespread adoption. The study demonstrates that a compact panel of 27 structured clinical, neuropsychological, and volumetric variables can match—or even exceed—the performance of MRI-based models when applied to both ML and DL methods.

In binary classification tasks, GB consistently outperformed all other models. For the CN–EMCI comparison, GB achieved a mean accuracy of 92% under 10-fold cross-validation when classes were balanced with SMOTE, and it retained similar margins in the CN–LMCI and LMCI–EMCI tasks. However, generating synthetic instances caused precision and recall to decline, indicating mild overfitting (Table 4). Conversely, the SVC yielded more balanced metrics and lower standard deviations in highly imbalanced settings, suggesting that artificial balancing can introduce artifacts—particularly in the CN class. Discriminating LMCI from AD proved especially difficult because the two stages are clinically similar. Although CNN + DigitCapsule-Net achieved the highest accuracy, every model displayed some degree of overfitting, reflecting the scarcity of authentic LMCI and AD examples in this comparison.

These trends persisted in the multiclass experiments. In three-class scenarios such as AD–CN–LMCI or AD–CN–EMCI, the addition of synthetic data *via* SMOTE stabilized performance and outperformed other resampling techniques (Tables 5 and 6);

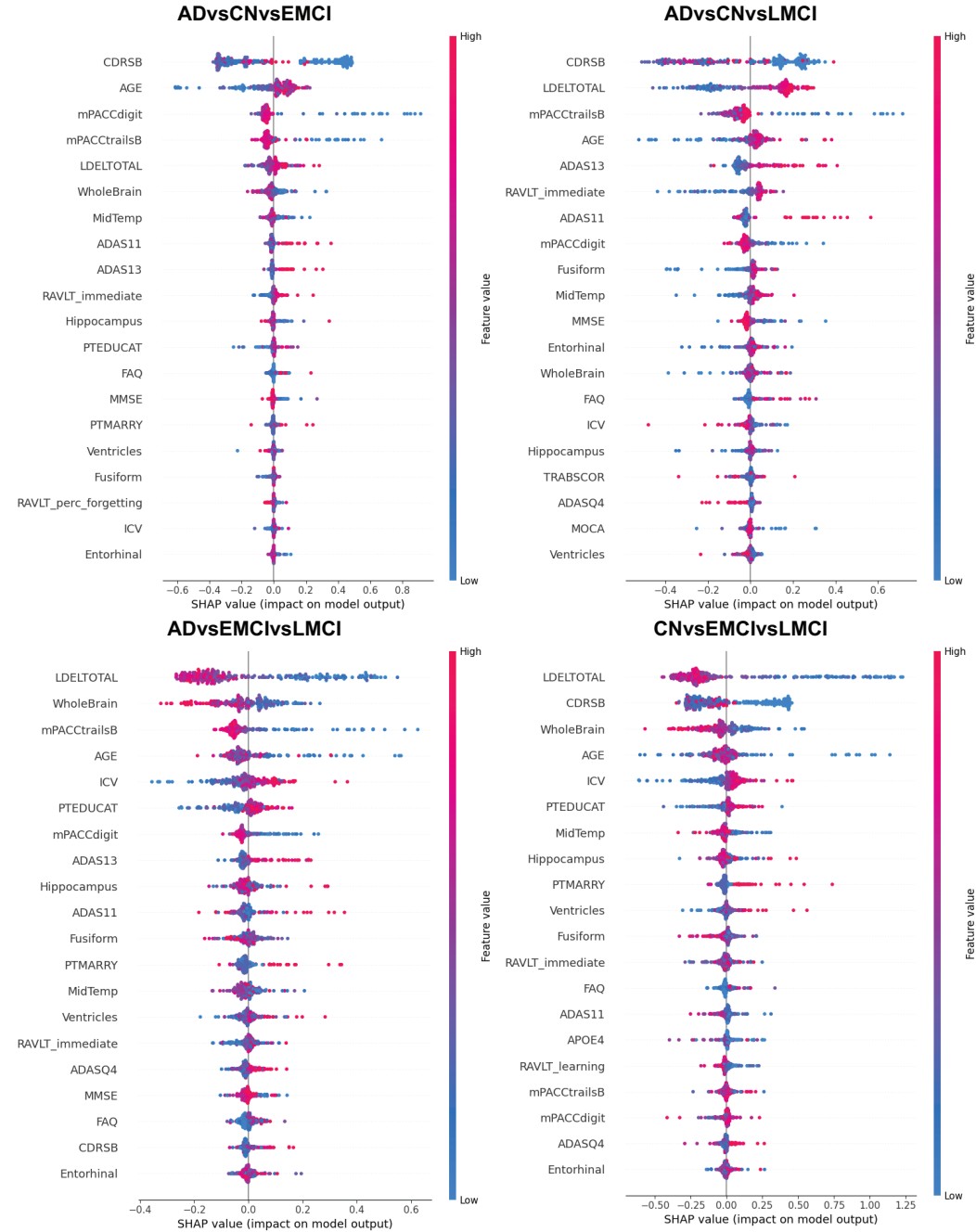

**Figure 17 Shap values plot obtained from the GB model for each class with imbalanced data.** The y-axis represents each of the 27 features, each number of which is specified in the Table 1.

nevertheless, SVC remained the algorithm least prone to overfitting under extreme imbalance. For AD–CN–LMCI, the CNN + DigitCapsule-Net yielded high accuracy with low variance, whereas in AD–CN–EMCI, GB once again delivered the best performance. In the four-class task (AD–CN–EMCI–LMCI), the CNN + DigitCapsule-Net again demonstrated robustness (Table 7), with SVC showing comparable stability. Nevertheless,

**Table 11 Comparison of classification accuracy between the state-of-the-art XGBoost-SHAP model and the proposed CNN+DigitCapsule-Net architecture for the three-class classification task (AD *vs* EMCI *vs* CN).**

| Model | Classes | Accuracy [%] |
|---|---|---|
| XGBoost-SHAP [14] | AD *vs* EMCI *vs* CN | 87.00 |
| Proposed (CNN+DigitCapsule-Net) | AD *vs* EMCI *vs* CN | 90.58 |

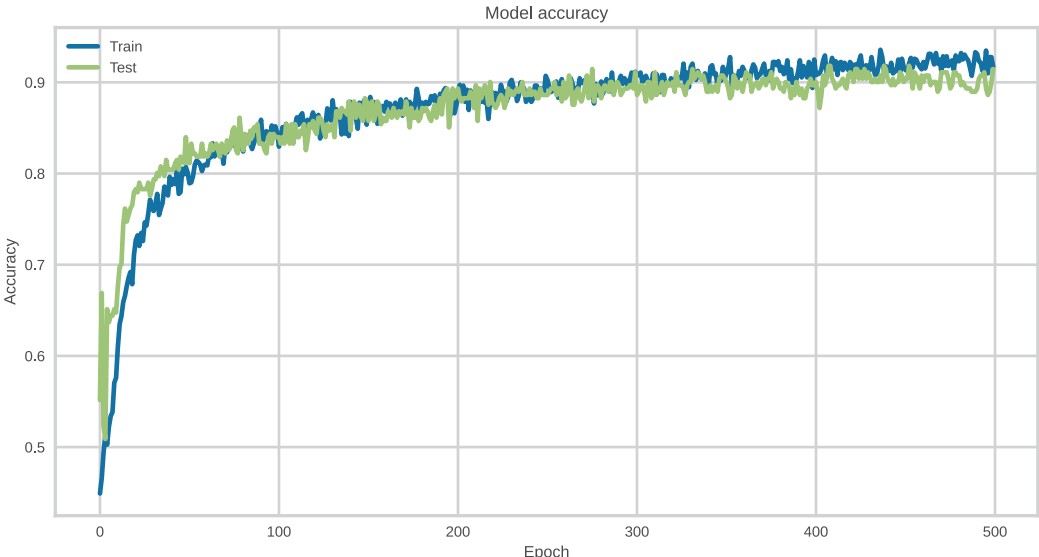

**Figure 18 Training and test accuracy curves for the CNN + DigitCapsule-Net model during the classification of AD,EMCI, and CN subjects.** The model converges steadily with no signs of over-fitting, achieving a final test accuracy of 90.58%.

layer-wise correlation analysis of the CNN revealed diminishing signal similarity with increasing depth, which hampered differentiation between EMCI and LMCI. Grad-CAM visualizations (Fig. 15) confirmed this convergence of deep-layer activations and elucidated the bottlenecks observed in those classifications.

SHAP values interpretability analysis (Fig. 17) consistently ranked CDRSB, LDELTOTAL, mPACCtrailsB, AGE, and WholeBrain as the most influential variables. Higher CDRSB scores and advanced age increased the likelihood of AD, while greater brain volume, especially WholeBrain, reduced it. These insights are readily interpretable for clinicians, whereas CNN activation maps—although visually informative—lack explicit decision rules, complicating clinical validation.

The practical advantages of ML over DL emerge across several dimensions. First, the sample-to-parameter ratio is more favorable for ensemble methods. With 1,846 patients and 27 features, GB trains only tens of thousands of parameters, whereas CNN + TF or CNN + DigitCapsule-Net exceed one million, leading to chronic underfitting and higher fold-to-fold variance. Second, variable relationships—among screening scores, cognitive thresholds, and volumetric markers—are largely linear or threshold-based; ensemble methods capture these patterns effectively without requiring hierarchical representations.

Third, implicit regularization—pruning and shrinkage in GB act as natural complexity controls, whereas DL requires additional mechanisms, such as dropout, batch normalization, and early stopping, which still fail to prevent overfitting, particularly with synthetic data. Fourth, computational cost—GB converges within minutes on a central processing unit (CPU), whereas CNN+DigitCapsule-Net requires several graphic processing unit (GPU) hours for 500 epochs, posing a substantial logistical barrier in resource-limited clinical environments. Finally, interpretability—whereas tree-based decision paths can be visualized and validated with SHAP, DL activations remain opaque and less conducive to regulatory compliance and clinical scrutiny.

However, using clinical data alone introduces notable limitations. First, anatomical granularity is compromised; although volumetric biomarkers like WholeBrain and Hippocampus are preserved, the detailed spatial information inherent to MRI is no longer available. Second, single-source bias—clinical measurements vary across evaluators and are influenced by psychosocial factors; incorporating longitudinal data or digital biomarkers (*e.g.*, wearable devices) could help mitigate intra-subject noise. Third, expert dependency—extensive neuropsychological testing requires trained personnel and may be more time-consuming than a single MRI acquisition in advanced clinical settings. Fourth, demographic generalizability—variables such as APOE4 status or educational level may behave differently across cohorts, and the lack of imaging data limits the ability to adjust for these biases. Consequently, future studies should pursue multimodal integration strategies that combine clinical, MRI, PET, or EEG data while upholding principles of transparency, reproducibility, and explainability.

Regarding resampling techniques such as SMOTE and ADASYN, while they improved average accuracy and reduced variance, they also increased sensitivity to noise and promoted overfitting, particularly in classes with overlapping boundaries, such as EMCI and LMCI. By focusing on borderline regions, ADASYN exacerbated sensitivity to noise. Nonparametric statistical tests subsequently confirmed that no significant differences existed among models or class-balancing techniques. Friedman's test yielded $\chi^2 = 2.89$ ($p = 0.23$) for AD–CN–EMCI and $\chi^2 = 1.12$ ($p = 0.57$) for CN–EMCI–LMCI. Wilcoxon pairwise comparisons showed ($p > 0.79$) for GB *vs* ETC and GB *vs* RF in several tasks. Levene's test indicated homogeneous variances ($p > 0.4$) across conditions, except ADASYN experiments, which exhibited additional variability. These results emphasize that the net benefit depends more on the quality of the synthetic data than on the specific resampling method, highlighting the need for rigorous validation procedures when altering original distributions.

This study addresses two critical gaps in the literature. First, it provides the first systematic and reproducible comparison between ML and DL applied to structured clinical data with open-source code and datasets. Second, it quantitatively assesses the statistical impact of multiple resampling strategies in AD multiclass scenarios. The findings challenge the widespread assumption of DL's automatic superiority and demonstrate that, under certain conditions of dimensionality and sample size, ML models not only match but surpass DL counterparts, with clear advantages in interpretability and computational efficiency.

Furthermore, the impact of this study is underscored by the limited number of related works in the field. While the referenced comparative study carried out several experiments, it reported only one set of results, which our proposed DigitCapsule-Net model outperformed, as shown in Table 11. The DigitCapsule-Net training and testing curve is shown in Fig. 18.

Finally, for AD stratification based exclusively on structured clinical data, ensemble methods (GB, RF, ETC) and margin-based models (SVC) offer the most favorable balance of accuracy, transparency, and computational efficiency. DL architectures may add value to tasks that involve complex hierarchical patterns or multimodal inputs (*e.g.*, MRI). However, their deployment must remain context-sensitive, balancing accuracy, interpretability, and resource constraints. In future work, integrating multimodal approaches and systematically applying robust statistical tests will be essential for validating apparent performance gains and advancing explainable-AI solutions in neurological contexts.

## CONCLUSION

This study presents a rigorous comparative analysis of traditional ML techniques and state-of-the-art DL architectures for AD, using exclusively structured clinical data from the ADNI dataset. A curated set of 27 features was selected through a hybrid feature-selection process that combined Boruta, Elastic Net, and information gain. The investigation showed that carefully optimized ML models—such as GB and SVC—combined with data-resampling techniques (*e.g.*, SMOTE) can outperform or match the accuracy of more complex DL architectures. These include a CNN + TF and a CNN + DigitCapsule-Net, yet require substantially lower computational resources.

Among DL models, the hybrid CNN + DigitCapsule-Net obtained the highest accuracy in several multi-class scenarios—particularly the AD–EMCI–CN task—achieving 90.58% and surpassing previously reported clinical-data models. However, these performance gains came at the cost of longer training times and reduced interpretability. SHAP values and Grad-CAM analyses showed thatCDRSB, LDELTOTAL, and mPACCtrailsB were consistently the most influential across models, reinforcing their clinical relevance.

From a methodological standpoint, statistical testing indicated that performance differences between models were not statistically significant in several comparisons. Nonparametric tests (Friedman and Wilcoxon) together with Levene's variance test confirmed the consistency and robustness of ML approaches across sampling and classification settings. The public availability of source code and curated datasets further strengthens the transparency and reproducibility of the results.

In summary, the evidence supports optimized ML models as practical, interpretable, and efficient alternatives for AD stratification when imaging modalities are unavailable or cost-prohibitive. This balance of accuracy, explainability, and computational feasibility renders these models well-suited for deployment in primary-care or remote clinical environments.

## FUTURE WORK

- Multimodal integration: Incorporate additional data modalities—such as MRI, PET, and EEG—alongside clinical features to exploit complementary patterns and enhance diagnostic accuracy. Explore multimodal-fusion strategies (early, late, or hybrid) within both ML and DL frameworks.

- Longitudinal modeling: Extend the analysis to longitudinal data to capture disease-progression dynamics. Apply recurrent neural architectures (*e.g.*, LSTM) or temporal Transformers to sequences of clinical measurements.

- Explainability enhancement: Integrate interpretability techniques beyond SHAP and Grad-CAM—including counterfactual analysis and concept-based explanations—to better support clinical decision-making and regulatory compliance.

- Generalizability assessment: Validate model performance on external datasets drawn from diverse demographic and clinical populations to assess robustness, especially in underserved or non-Western cohorts.

- Edge deployment and optimization: Investigate lightweight model variants or compression techniques (*e.g.*, pruning or quantization) to enable real-time inference in resource-constrained settings such as rural clinics or mobile-health platforms.

- Integration with EHR Systems: Integrate the developed models into electronic health-record (EHR) platforms to support automated screening and triage in routine-care workflows.

- Ethical and regulatory considerations: Conduct studies on algorithmic fairness, bias mitigation, and regulatory compliance to ensure equitable deployment and foster trust in AI-powered diagnostic tools.

- Explore alternative data processing methodologies to identify relevant features and analyze broader datasets.

### Funding

Mario Alejandro Bravo-Ortíz and Sergio Alejandro Holguin García are supported by a Ph.D. grant Convocatoria 22 OCAD de Ciencia, Tecnología e Innovación del Sistema General de Regalías de Colombia y Ministerio de Ciencia, Tecnología e Innovación de Colombia. The Universidad Autónoma de Manizales included this as part of the projects "Clasificación de los estadios del Alzheimer utilizando Imágenes de Resonancia Magnética Nuclear y datos clínicos a partir de técnicas de Deep Learning" (No. 873-139) and "Aplicación de Vision Transformer para clasificar estadios del Alzheimer utilizando imágenes de resonancia magnética nuclear y datos clínicos" (No. 847-2023 TD). This research was also funded by ANID FONDECYT (No. 1230315), ANID-MILENIO (No. NCN2024_103), ANID PIA/ BASAL (No. AFB240003), and ANID/PIA/ANILLO (No. ACT210096). This was also supported by the Universidad de Caldas (No. PRY-89) as part of the project "Plataforma tecnológica para la clasificación de los estadios de la enfermedad de Alzheimer utilizando

imágenes de resonancia magnética nuclear, datos clínicos y técnicas de deep learning." Additional funding was provided by the National Agency for Research and Development (ANID); Applied Research Subdirection (SIA) through the instrument IDeA I+D 2023 (No. ID23I10357) and the project "Interactive Virtual Didactic Strategy for the Promotion of ICT Skills and their Relationship with Computational Thinking" (No. PRY-121). Data collection and sharing for this project was funded by the Alzheimer's Disease Neuroimaging Initiative (ADNI) (National Institutes of Health Grant U01 AG024904) and DOD ADNI (Department of Defense award number W81XWH-12-2-0012). ADNI is funded by the National Institute on Aging, the National Institute of Biomedical Imaging and Bioengineering, and through generous contributions from the following: AbbVie, Alzheimer's Association; Alzheimer's Drug Discovery Foundation; Araclon Biotech; BioClinica, Inc.; Biogen; Bristol-Myers Squibb Company; CereSpir, Inc.; Cogstate; Eisai Inc.; Elan Pharmaceuticals, Inc.; Eli Lilly and Company; EuroImmun; F. Hoffmann-La Roche Ltd and its affiliated company Genentech, Inc.; Fujirebio; GE Healthcare; IXICO Ltd.; Janssen Alzheimer Immunotherapy Research & Development, LLC.; Johnson & Johnson Pharmaceutical Research & Development LLC.; Lumosity; Lundbeck; Merck & Co., Inc.; Meso Scale Diagnostics, LLC.; NeuroRx Research; Neurotrack Technologies; Novartis Pharmaceuticals Corporation; Pfizer Inc.; Piramal Imaging; Servier; Takeda Pharmaceutical Company; and Transition Therapeutics. The Canadian Institutes of Health Research is providing funds to support ADNI clinical sites in Canada. The funders had no role in study design, data collection and analysis, decision to publish, or preparation of the manuscript.

## Grant Disclosures

The following grant information was disclosed by the authors:
Tecnología e Innovación del Sistema General de Regalías de Colombia y Ministerio de Ciencia, Tecnología e Innovación de Colombia.
The Universidad Autónoma de Manizales: 873-139.
ANID FONDECYT: 1230315.
ANID-MILENIO: NCN2024_103.
ANID PIA/BASAL: AFB240003.
ANID/PIA/ANILLO: ACT210096.
Universidad de Caldas: PRY-89.
National Agency for Research and Development (ANID).
Applied Research Subdirection (SIA): ID23I10357.
National Institutes of Health Grant: U01 AG024904.
DOD ADNI (Department of Defense): W81XWH-12-2-0012.
National Institute on Aging.
National Institute of Biomedical Imaging and Bioengineering.
AbbVie.
Alzheimer's Association.
Alzheimer's Drug Discovery Foundation.
Araclon Biotech.
BioClinica, Inc.

Biogen.
Bristol-Myers Squibb Company.
CereSpir, Inc.
Cogstate.
Eisai Inc.
Elan Pharmaceuticals, Inc.
Eli Lilly and Company.
EuroImmun.
F. Hoffmann-La Roche Ltd.
Genentech, Inc.
Fujirebio.
GE Healthcare.
IXICO Ltd.
Janssen Alzheimer Immunotherapy Research & Development, LLC.
Johnson & Johnson Pharmaceutical Research & Development LLC.
Lumosity.
Lundbeck.
Merck & Co., Inc.
Meso Scale Diagnostics, LLC.
NeuroRx Research.
Neurotrack Technologies.
Novartis Pharmaceuticals Corporation.
Pfizer Inc.
Piramal Imaging.
Servier.
Takeda Pharmaceutical Company.
Transition Therapeutics.
Canadian Institutes of Health Research.

## Competing Interests

The authors declare that they have no competing interests.

## Author Contributions

- Mario Alejandro Bravo-Ortíz conceived and designed the experiments, performed the experiments, analyzed the data, performed the computation work, authored or reviewed drafts of the article, and approved the final draft.
- Sergio Alejandro Holguin-Garcia conceived and designed the experiments, performed the experiments, analyzed the data, performed the computation work, prepared figures and/or tables, authored or reviewed drafts of the article, and approved the final draft.
- Ernesto Guevara-Navarro performed the experiments, authored or reviewed drafts of the article, and approved the final draft.

- Esteban Cerón-Cabrera performed the experiments, authored or reviewed drafts of the article, and approved the final draft.
- Alejandro Mora-Rubio performed the experiments, prepared figures and/or tables, authored or reviewed drafts of the article, and approved the final draft.
- Harold Brayan Arteaga-Arteaga conceived and designed the experiments, prepared figures and/or tables, authored or reviewed drafts of the article, and approved the final draft.
- Gonzalo A. Ruz performed the computation work, authored or reviewed drafts of the article, and approved the final draft.
- Reinel Tabares-Soto performed the computation work, authored or reviewed drafts of the article, and approved the final draft.

## Data Availability

The data is available from ADNI. To procure the tabular data up to the ADNI3 phase, a formal application was submitted *via* the Alzheimer's Disease Neuroimaging Initiative (ADNI) website (adni. loni.usc.edu). This process involved the completion of an online application form and the acceptance of the Data Use Agreement. Upon approval of the application by the ADNI Data Sharing and Publications Committee, access credentials for the Laboratory of Neuro Imaging (LONI) Image and Data Archive (IDA) were provided. The required tabular datasets were then accessed by logging into the IDA.Within the download section, under "Study Data," the comprehensive ADNIMERGE file, containing merged data from ADNI1, ADNI GO, ADNI2, and ADNI3, was selected and subsequently downloaded in comma-separated values (CSV) format for analysis.

The code to reproduce the analyses and scripts are available at Zenodo: Mario Alejandro Bravo-Ortiz. (2025). MarioBravo12/Alzheimer-s-Disease-Classification-Using-Clinical-Data1: Creating citation file (v1.0.2). Zenodo. https://doi.org/10.5281/zenodo.15860467.

## Supplemental Information

Supplemental information for this article can be found online at http://dx.doi.org/10.7717/peerj-cs.3208#supplemental-information.

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
