# Peer review of "Transformers and capsule networks *vs* classical ML on clinical data for alzheimer classification"

_PeerJ Computer Science, doi:10.7717/peerj-cs.3208_

## Round 0.1 · original submission · Major Revisions

· Academic Editor

Major Revisions

The reviewers have substantial concerns about this manuscript. The authors should provide point-to-point responses to address all the concerns and provide a revised manuscript with the revised parts being marked in different color.

**Language Note:** The review process has identified that the English language must be improved. PeerJ can provide language editing services - please contact us at [email protected] for pricing (be sure to provide your manuscript number and title). Alternatively, you should make your own arrangements to improve the language quality and provide details in your response letter. – PeerJ Staff

Reviewer 1 ·

Basic reporting

Improve English clarity and abstract.
Add statistical significance testing of results.
Explain model architectures and the training process in more detail.

Experimental design

Improve section organization and figure references.
Discuss overfitting and model generalization more explicitly.
Visualize feature importance and include ablation studies.

Validity of the findings

Include stratified k-fold cross-validation results if available.
Add a brief model interpretability discussion per model type.

Additional comments

Please see attached file report!

Annotated reviews are not available for download in order to protect the identity of reviewers who chose to remain anonymous.

Reviewer 2 ·

Basic reporting

- English language usage needs revision. Numerous instances of awkward phrasing, minor grammatical errors, and typographical issues (e.g., “classiûcation” instead of “classification”, “Trasnformer” instead of “Transformer”) detract from the clarity. A thorough language edit by a native speaker or a professional editing service is highly recommended.

- The text occasionally presents long, complex sentences that could be broken up for better readability. Consider simplifying sentence structure where possible.

- The introduction provides a reasonable background on Alzheimer’s disease and the challenges of diagnosis, as well as a discussion of machine learning and deep learning approaches. However, while the context is set, the motivation for comparing specific models (e.g., CNN+DigitCapsule-Net, CNN+Transformer Encoder, traditional ML models) could be strengthened. Explicitly state the knowledge gap and the potential impact of the comparative study.

- There are several formatting inconsistencies (e.g., the placement of figures and tables, numbering of sections) that should be aligned with PeerJ’s standards.

- Section headings and subheadings are generally clear, but it would help to ensure that each section begins with a concise statement of its purpose.

- Overall, the figures are relevant to the content, but several images suffer from quality issues (low resolution or unclear labeling). It is strongly recommended that all figures be produced at high resolution and that axis labels, legends, and annotations abelegible in both digital and print formats.

- Also, consider revising the color schemes of the plots to ensure that they are accessible (e.g., for color-blind readers) and conform to journal standards.

Experimental design

- The research question is well-defined: to compare the performance of advanced deep learning models (including a Transformer Encoder and DigitCapsule-Net) with traditional ML methods for classifying Alzheimer’s disease stages using clinical data. The rationale for focusing on clinical data, as opposed to more commonly used imaging data, is clearly stated and represents a novel approach. As per the ADNI database, use a proper way to reference it as a link or cite a paper by the database providers.

- The methods section is comprehensive, describing data acquisition from the ADNI database, data pre-processing (including feature selection and handling missing values), and the application of several oversampling techniques (SMOTE, ADASYN, etc.) to balance the dataset. As well, the description of each machine learning and deep learning model is detailed, including the mathematical formulations (e.g., for SVM, gradient boosting, CNN, and Capsule-Net). This level of detail is commendable, but some parts could be streamlined to improve readability.

- As per the use of oversampling methods, consider and discuss clearly over different aspects: 1) SMOTE and ADASYN may not be advisable in situations involving high-dimensional data, noisy or mislabeled instances, significant class overlap, very small datasets, time series or sequential data, and datasets dominated by categorical variables. 2) In such cases, these techniques can generate unrealistic or noisy synthetic samples, amplify class confusion, or lead to overfitting. 3) Alternative strategies like ensemble methods, cost-sensitive learning, anomaly detection, or domain-specific data augmentation may be more suitable depending on the context.

- The manuscript provides sufficient details about hyperparameters, model architectures, and the experimental pipeline (including the use of grid search and cross-validation). However, additional clarity on the random seed settings and software versions (beyond the brief mention of Google Colab specifications) would be beneficial for replication.

- The use of multiple evaluation metrics (accuracy, precision, recall, F1 score, confusion matrices, and ROC curves) demonstrates a rigorous approach to model validation. It would also be helpful to include statistical tests to compare model performances formally. For example, provide p-values or confidence intervals when claiming superiority of one model over another.

Validity of the findings

- The results section presents an extensive evaluation of the models on both balanced and unbalanced datasets. The inclusion of cross-validation helps to assess the robustness of the findings. The discussion of results is mostly quantitative. However, the manuscript would benefit from a more in-depth qualitative interpretation. Specifically, the implications of the differences between deep learning and traditional ML models should be further explored, including potential limitations of using clinical data exclusively.

- The comparative analysis is a strong aspect of the study. The novel incorporation of Transformer Encoders and Capsule Networks in this context is innovative. Nonetheless, the authors should discuss potential reasons why traditional models might still outperform advanced deep learning models in certain scenarios with structured clinical data, and consider including a discussion of computational efficiency or interpretability.

- While the study is comprehensive, the authors should acknowledge limitations more clearly. For example, the generalizability of the findings may be limited by the dataset size or the inherent biases in clinical data.

- The manuscript follows PeerJ’s guidelines on data sharing and reproducibility. Ensure that all raw data and code are accessible in the final published version.

Additional comments

The following is a general assessment of the manuscript: The paper addresses a timely and important challenge—enhancing early Alzheimer’s disease diagnosis through machine learning, offering a comprehensive methodological framework and a valuable comparison between deep learning and traditional techniques. However, the manuscript requires extensive language editing to improve clarity, along with revisions to several figures for better quality and accessibility. Some method sections, particularly those with complex mathematical formulations, could be simplified or moved to supplementary materials to maintain narrative flow. The discussion should be expanded to better interpret the results, highlight clinical implications, and address limitations. Additionally, consistency in terminology and notation throughout the text needs to be ensured.

---

## Round 0.2 · Major Revisions

· Academic Editor

Major Revisions

There are some major concerns that need to be addressed.

Reviewer 1 ·

Basic reporting

small LMCI sample, generalization to external datasets

Experimental design

Please change the title of paper. Make it short and remain main idea.

Validity of the findings

The manuscript is well-structured and written in clear scientific English.

Annotated reviews are not available for download in order to protect the identity of reviewers who chose to remain anonymous.

Reviewer 2 ·

Basic reporting

No comment.

Experimental design

No comment.

Validity of the findings

No comment.

Additional comments

The authors have properly addressed all my concerns. Looking forward to trying the last version of your code.

---

## Round 0.3 · accepted · Accept

· Academic Editor

Accept

The reviewers have addressed the remaining concerns. I recommend accepting this manuscript.